# Learning Neural Implicit through Volume Rendering with Attentive Depth Fusion Priors

**Pengchong Hu**     **Zhizhong Han**
Machine Perception Lab, Wayne State University, Detroit, USA
pchu@wayne.edu     h312h@wayne.edu

## Abstract

Learning neural implicit representations has achieved remarkable performance in 3D reconstruction from multi-view images. Current methods use volume rendering to render implicit representations into either RGB or depth images that are supervised by multi-view ground truth. However, rendering a view each time suffers from incomplete depth at holes and unawareness of occluded structures from the depth supervision, which severely affects the accuracy of geometry inference via volume rendering. To resolve this issue, we propose to learn neural implicit representations from multi-view RGBD images through volume rendering with an attentive depth fusion prior. Our prior allows neural networks to perceive coarse 3D structures from the Truncated Signed Distance Function (TSDF) fused from all depth images available for rendering. The TSDF enables accessing the missing depth at holes on one depth image and the occluded parts that are invisible from the current view. By introducing a novel attention mechanism, we allow neural networks to directly use the depth fusion prior with the inferred occupancy as the learned implicit function. Our attention mechanism works with either a one-time fused TSDF that represents a whole scene or an incrementally fused TSDF that represents a partial scene in the context of Simultaneous Localization and Mapping (SLAM). Our evaluations on widely used benchmarks including synthetic and real-world scans show our superiority over the latest neural implicit methods. Please see our project page for code and data at https://machineperceptionlab.github.io/Attentive_DF_Prior/.

## 1   Introduction

3D reconstruction from multi-view images has been studied for decades. Traditional methods like Structure from Motion (SfM) [61], Multi-View Stereo (MVS) [62], and Simultaneous Localization and Mapping (SLAM) [49] estimate 3D structures as point clouds by maximizing multi-view color consistency. Current methods [84, 28] mainly adopt data-driven strategies to learn depth estimation priors from large scale benchmarks using deep learning models. However, the reconstructed 3D point clouds lack geometry details, due to their discrete representation essence, which makes them not friendly to downstream applications.

More recent methods use implicit functions such as signed distance functions (SDFs) [77] or occupancy functions [51] as continuous representations of 3D shapes and scenes. Using volume rendering, we can learn neural implicit functions by comparing their 2D renderings with multi-view ground truth including color [88, 97], depth [88, 3, 97] or normal [88, 76, 16] maps. Although the supervision of using depth images as rendering target can provide detailed structure information and guide importance sampling along rays [88], both the missing depth at holes and the unawareness of occluded structures make it hard to significantly improve the reconstruction accuracy. Hence, how to more

effectively leverage depth supervision for geometry inference through volume rendering is still a challenge.

To overcome this challenge, we introduce to learn neural implicit through volume rendering with an attentive depth fusion prior. Our key idea is to provide neural networks the flexibility of choosing geometric clues, i.e., the geometry that has been learned and the Truncated Signed Distance Function (TSDF) fused from all available depth images, and combining them into neural implicit for volume rendering. We regard the TSDF as a prior sense of the scene, and enable neural networks to directly use it as a more accurate representation. The TSDF enables accessing the missing depth at holes on one depth image and the occluded structures that are invisible from the current view, which remedies the demerits of using depth ground truth to supervise rendering. To achieve this, we introduce an attention mechanism to allow neural networks to balance the contributions of currently learned geometry and the TSDF in the neural implicit, which leads the TSDF into an attentive depth fusion prior. Our method works with either known camera poses or camera tracking in the context of SLAM, where our prior could be either a one-time fused TSDF that represents a whole scene or an incrementally fused TSDF that represents a partial scene. We evaluate our performance on benchmarks containing synthetic and real-world scans, and report our superiority over the latest methods with known or estimated camera poses. Our contributions are listed below.

i) We present a novel volume rendering framework to learn neural implicit representations from RGBD images. We enable neural networks to use either currently learned geometry or the one from depth fusion in volume rendering, which leads to a novel attentive depth fusion prior for learning neural implicit functions inheriting the merits of both the TSDF and the inference.

ii) We introduce a novel attention mechanism and a neural network architecture to learn attention weights for the attentive depth fusion prior in neural rendering with either known camera poses or camera pose tracking in SLAM.

iii) We report the state-of-the-art performance in surface reconstruction and camera tracking on benchmarks containing synthetic and real-world scans.

## 2 Related Work

Learning 3D implicit functions using neural networks has made huge progress [61, 62, 46, 51, 88, 78, 73, 76, 16, 56, 34, 25, 11, 93, 42, 44, 9, 43, 4, 33, 5, 19, 26, 47, 20, 70, 12, 48, 55, 17, 27, 79, 35, 32, 21, 7, 58, 86, 52]. We can learn neural implicit functions from 3D ground truth [23, 8, 54, 45, 68, 39, 69], 3D point clouds [94, 41, 15, 1, 92, 2, 10] or multi-view images [46, 14, 51, 77, 88, 78, 73, 76, 16]. We briefly review methods with multi-view supervision below.

### 2.1 Multi-view Stereo

Classic multi-view stereo (MVS) [61, 62] employs multi-view photo consistency to estimate depth maps from multiple RGB images. They rely on matching key points on different views, and are limited by large viewpoint variations and complex illumination. Without color, space carving [30] is also an effective way of reconstructing 3D structure as voxel grids.

Recent methods employ data driven strategy to train neural networks to predict depth maps from either depth supervision [84] or multi-view photo consistency [95].

### 2.2 Neural Implicit from Multi-view Supervision

Early works leverage various differentiable renderers [63, 38, 26, 89, 37, 80, 50, 36, 67] to render the learned implicit functions into images, so that we can measure the error between rendered images and ground truth images. These methods usually require masks to highlight the object, and use surface rendering to infer the geometry, which limits their applications in scenes. Similarly, DVR [50] and IDR [87] predict the radiance near surfaces.

With volume rendering, NeRF [46] and its variations [53, 47, 57, 60, 97, 3, 75, 6, 98, 67] model geometry and color together. They can generate plausible images from novel viewpoints, and do not need masks during rendering procedure. UNISURF [51] and NeuS [77] learn occupancy functions and SDFs by rendering them with colors using revised rendering equations. Following methods

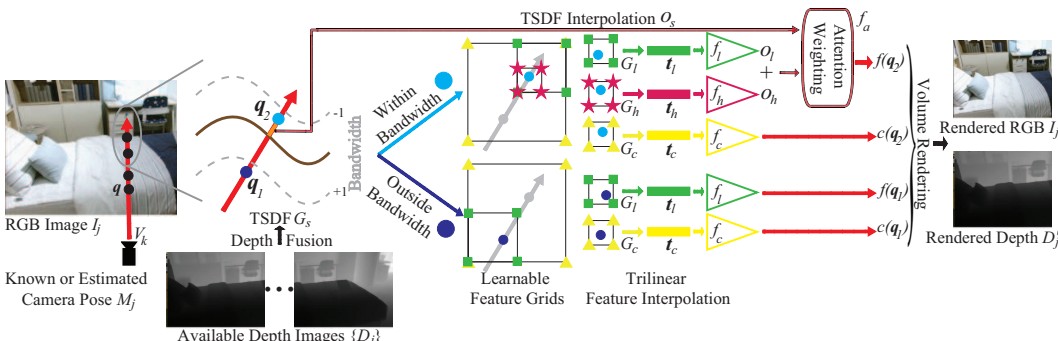

Figure 1: Overview of our method.

improve accuracy of implicit functions using additional priors or losses related to depth [88, 3, 97], normals [88, 76, 16], and multi-view consistency [14].

Depth images are also helpful to improve the inference accuracy. Depth information can guide sampling along rays [88] or provide rendering supervision [88, 3, 97, 96, 24, 31, 82, 98], which helps neural networks to estimate surfaces.

### 2.3  Neural Implicit with SLAM

Given RGBD images, more recent methods [91, 81, 72, 59] employ neural implicit representations in SLAM. iMAP [66] shows that an MLP can serve as the only scene representation in a realtime SLAM system. NICE-SLAM [97] introduces a hierarchical scene representation to reconstruct large scenes with more details. NICER-SLAM [96] uses easy-to-obtain monocular geometric cues without requiring depth supervision. Co-SLAM [74] jointly uses coordinate and sparse parametric encodings to learn neural implicit functions. Segmentation priors [29, 18] show their potentials to improve the performance of SLAM systems. With segmentation priors, vMAP [29] represents each object in the scene as a neural implicit in a SLAM system.

Instead of using depth ground truth to only supervise rendering, which contains incomplete depth and unawareness of occlusion, we allow our neural network to directly use depth fusion priors as a part of neural implicit, and determine where and how to use depth priors along with learned geometry.

## 3  Method

**Overview.** We present an overview of our method in Fig. 1. Given RGBD images $\{I_j, D_j\}_{j=1}^J$ from $J$ view angles, where $I_j$ and $D_j$ denote the RGB and depth images, available either all together or in a streaming way, we aim to infer the geometry of the scene as an occupancy function $f$ which predicts the occupancy $f(\boldsymbol{q})$ at arbitrary locations $\boldsymbol{q} = (x, y, z)$. Our method works with camera poses $\{M_j\}$ that are either known or estimated by our camera tracking method in the context of SLAM.

We learn the occupancy function through volume rendering using the RGBD images $\{I_j, D_j\}_{j=1}^J$ as supervision. We render $f$ with a color function $c$ which predicts an RGB color $c(\boldsymbol{q})$ at locations $\boldsymbol{q}$ into an RGB image $I_j'$ and a depth image $D_j'$, which are optimized to minimize their rendering errors to the supervision $\{I_j, D_j\}$.

We start from shooting rays $\{V_k\}$ from current view $I_j$, and sample queries $\boldsymbol{q}$ along each ray $V_k$. For each query $\boldsymbol{q}$, we employ learnable feature grids $G_l$, $G_h$, and $G_c$ covering the scene to interpolate its hierarchical geometry features $\boldsymbol{t}_l$, $\boldsymbol{t}_h$, and a color feature $\boldsymbol{t}_c$ by trilinear interpolation. Each feature is further transformed into an occupancy probability or RGB color by their corresponding decoders $f_l$, $f_h$, and $f_c$. For queries $\boldsymbol{q}$ inside the bandwidth of a TSDF grid $G_s$ fused from available depth images $\{D_i\}$, we leverage its interpolation from $G_s$ as a prior of coarse occupancy estimation. The prior is attentive by a neural function $f_a$ which determines the occupancy function $f(\boldsymbol{q})$ by combining currently learned geometry and the coarse estimation using the learned attention weights from $f_a$.

Lastly, we use the occupancy function $f(\boldsymbol{q})$ and the color function $c(\boldsymbol{q})$ to render $I'$ and $D'$ through volume rendering. With a learned $f$, we run the marching cubes [40] to reconstruct a surface.

**Depth Fusion and Bandwidth Awareness.** With known or estimated camera poses $\{M_j\}$, we get a TSDF by fusing depth images $\{D_j\}$ that are available either all together or in a streaming way. Our method can work with a TSDF that describes either a whole scene or just a part of the scene, and we will report the performance with different settings in ablation studies. We use the TSDF to provide a coarse occupancy estimation which removes the limit of the missing depth or the unawareness of occlusion on single depth supervision. The TSDF is a grid which predicts signed distances at any locations inside it through trilinear interpolation. The predicted signed distances are truncated with a threshold $\mu$ into a range of $[-1, 1]$ which covers a band area on both sides of the surface in Fig. 1.

Since the TSDF predicts signed distances for queries within the bandwidth with higher confidence than the ones outside the bandwidth, we only use the depth fusion prior within the bandwidth. This leads to different ways of learning geometries, as shown in Fig. 1. Specifically, for queries $q$ within the bandwidth, we model the low-frequency surfaces using a low resolution feature grid $G_l$, learn the high-frequency details as a complementation using a high resolution feature grid $G_h$, and let our attention mechanism to determine how to use the depth fusion prior from the TSDF $G_s$. Instead, we only use the low resolution feature grid $G_l$ to model densities outside the bandwidth during volume rendering. Regarding color modeling, we use the feature grid $G_c$ with the same size for interpolating color of queries over the scene.

**Feature Interpolation.** For queries $q$ sampled along rays, we use the trilinear interpolation to obtain its features $\boldsymbol{h}_l$, $\boldsymbol{h}_h$, and $\boldsymbol{h}_c$ from learnable $G_l$, $G_h$, $G_c$ and occupancy estimation $o_s$ from $G_s$. Both feature grids and the TSDF grid cover the whole scene and use different resolutions, where learnable feature vectors are associated with vertices on each feature grid. Signed distances at vertices on $G_s$ may change if we incrementally fuse depth images in the context of SLAM.

**Occupancy Prediction Priors.** Similar to NICE-SLAM [97], we use pre-trained decoders $f_l$ and $f_h$ to predict low frequency occupancies $o_l$ and high frequency ones $o_h$ from the interpolated features $\boldsymbol{t}_l$ and $\boldsymbol{t}_h$, respectively. We use $f_l$ and $f_h$ as an MLP decoder in ConvONet [54] respectively, and minimize the binary cross-entropy loss to fit the ground truth. After pre-training, we fix the parameters in $f_l$ and $f_h$, and use them to predict occupancies below,

$$o_l = f_l(\boldsymbol{q}, \boldsymbol{t}_l), \ o_h = f_h(\boldsymbol{q}, \boldsymbol{t}_h), \ o_{lh} = o_l + o_h, \tag{1}$$

where we enhance $f_h$ by concatenating $f_l$ as $f_h \leftarrow [f_h, f_l]$ and denote $o_{lh}$ as the occupancy predicted at query $q$ by the learned geometry.

**Color Predictions.** Similarly, we use the interpolated feature $\boldsymbol{t}_c$ and an MLP decoder $f_c$ to predict color at query $q$, i.e., $c(\boldsymbol{q}) = f_c(\boldsymbol{q}, \boldsymbol{t}_c)$. We predict color to render RGB images for geometry inference or camera tracking in SLAM. The decoder is parameterized by parameters $\boldsymbol{\theta}$ which are optimized with other learnable parameters in the feature grids.

**Attentive Depth Fusion Prior.** We introduce an attention mechanism to leverage the depth fusion prior. We use a deep neural network to learn attention weights to allow networks to determine how to use the prior in volume rendering.

As shown in Fig. 1, we interpolate a signed distance $s \in [-1, 1]$ at query $q$ from the TSDF $G_s$ using trilinear interpolation, where $G_s$ is fused from available depth images. We formulate this interpolation as $s = f_s(\boldsymbol{q}) \in [-1, 1]$. We normalize $s$ into an occupancy $o_s \in [0, 1]$, and regard $o_s$ as the occupancy predicted at query $q$ by the depth fusion prior.

For query $q$ within the bandwidth, our attention mechanism trains an MLP $f_a$ to learn attention weights $\alpha$ and $\beta$ to aggregate the occupancy $o_{lh}$ predicted by the learned geometry and the occupancy $o_s$ predicted by the depth fusion prior, which leads the TSDF to an attentive depth fusion prior. Hence, our attention mechanism can be formulated as,

$$[\alpha, \beta] = f_a(o_{lh}, o_s), \text{ and } \alpha + \beta = 1. \tag{2}$$

We implement $f_a$ using an MLP with 6 layers. The reason why we do not use a complex network like Transformer is that we want to justify the effectiveness of our idea without taking too much credit from complex neural networks. Regarding the design, we do not use coordinates or positional encoding as a part of input to avoid noisy artifacts. Moreover, we leverage a Softmax normalization layer to achieve $\alpha + \beta = 1$, and we do not predict only one parameter $\alpha$ and use $1 - \alpha$ as the second weight, which also degenerates the performance. We will justify these alternatives in experiments.

For query $\boldsymbol{q}$ outside the bandwidth, we predict the occupancy using the feature $\boldsymbol{t}_l$ interpolated from the low frequency grid $G_l$ and the decoder $f_l$ to describe the relatively simpler geometry. In summary, we eventually formulate our occupancy function $f$ as a piecewise function below,

$$f(\boldsymbol{q}) = \begin{cases} \alpha \times o_{lh} + \beta \times o_s, & f_s(\boldsymbol{q}) \in (-1, 1) \\ o_l, & f_s(\boldsymbol{q}) = 1 \text{ or } -1 \end{cases} \quad (3)$$

**Volume Rendering.** We render the color function $c$ and occupancy function $f$ into RGB $I'$ and depth $D'$ images to compare with the RGBD supervision $\{I, D\}$.

With camera poses $M_j$, we shoot a ray $V_k$ from view $I_j$. $V_k$ starts from the camera origin $\boldsymbol{m}$ and points a direction of $\boldsymbol{r}$. We sample $N$ points along the ray $V_k$ using stratified sampling and uniformly sampling near the depth, where each point is sampled at $\boldsymbol{q}_n = \boldsymbol{m} + d_n \boldsymbol{r}$ and $d_n$ corresponds to the depth value of $\boldsymbol{q}_n$ on the ray. Following UNISURF [51], we transform occupancies $f(\boldsymbol{q}_n)$ into weights $w_n$ which is used for color and depth accumulation along the ray $V_k$ in volume rendering,

$$w_n = f(\boldsymbol{q}_n) \prod_{n'=1}^{n-1} (1 - f(\boldsymbol{q}_{n'})), \ I(k)' = \sum_{n'=1}^{N} w_{n'} \times c(\boldsymbol{q}_{n'}), \ D(k)' = \sum_{n'=1}^{N} w_{n'} \times d_{n'}. \quad (4)$$

**Loss.** With known camera pose $M_j$, we render the scene into the color and depth images at randomly sampled $K$ pixels on the $j$-th view, and optimize parameters by minimizing the rendering errors,

$$L_I = \frac{1}{JK} \sum_{j,k=1}^{J,K} ||I_j(k) - I'_j(k)||_1, L_D = \frac{1}{JK} \sum_{j,k=1}^{J,K} ||D_j(k) - D'_j(k)||_1, \min_{\boldsymbol{\theta}, G_l, G_h, G_c} L_D + \lambda L_I. \quad (5)$$

**In the Context of SLAM.** We can jointly do camera tracking and learning neural implicit from streaming RGBD images. To achieve this, we regard the camera extrinsic matrix $M_j$ as learnable parameters, and optimize them by minimizing our rendering errors. Here, we follow [97] to weight the depth rendering loss to decrease the importance at pixels with large depth variance along the ray,

$$\min_{M_j} \frac{1}{JK} \sum_{j,k=1}^{J,K} \frac{1}{Var(D'_j(k))} ||D_j(k) - D'_j(k)||_1 + \lambda_1 L_I, \quad (6)$$

where $Var(D'_j(k)) = \sum_{n=1}^{N} w_n (D'_j(k) - d_n)^2$. Moreover, with streaming RGBD images, we incrementally fuse the most current depth image into TSDF $G_s$. Specifically, the incremental fusion includes a pre-fusion and an after-fusion stage. The pre-fusion aims to use a camera pose coarsely estimated by a traditional method to fuse a depth image onto a current TSDF to calculate a rendering error for a more accurate pose estimation at current frame. The after-fusion stage will refuse the depth image onto the current TSDF for camera tracking at the next frame. Please refer to our supplementary materials for more details.

We do tracking and mapping iteratively. For mapping procedure, we render $E$ frames each time and back propagate rendering errors to update parameters. $E$ frames include the current frame and $E - 1$ key frames that have overlaps with the current frame. For simplicity, we merely maintain a key frame list by adding incoming frames with an interval of 50 frames for fair comparisons.

**Details.** The optimization is performed at three stages, which makes optimization converge better. We first minimize $L_D$ by optimizing low frequency feature grid $G_l$, and then both low and high frequency feature grid $G_l$ and $G_h$, and finally minimize Eq 5 by optimizing $G_l$, $G_h$, and $G_c$. Our bandwidth from the TSDF $G_s$ covers 5 voxels on both sides of the surface. We shoot $K = 1000$ or $5000$ rays for reconstruction or tracking from each view, and render $E = 5$ or $10$ frames each time for fair comparison with other methods. We set $\lambda = 0.2$, $\lambda_1 = 0.5$ in loss functions. We sample $N = 48$ points along each ray for rendering. More implementation details can be found in our supplementary materials.

## 4 Experiments and Analysis

### 4.1 Experimental Setup

**Datasets.** We report evaluations on both synthetic datasets and real scans including Replica [64] and ScanNet [13]. For fair comparisons, we report results on the same scenes from Replica and ScanNet

Table 1: Reconstruction Comparisons on Replica.

| | COLMAP [62] | TSDF [90] | iMAP [66] | DI [22] | NICE [97] | Vox [83] | DROID [71] | NICER [96] | Ours | vMAP [29] | Ours* |
|---|---|---|---|---|---|---|---|---|---|---|---|
| **DepthL1** ↓ | - | 6.56 | 7.64 | 23.33 | 3.53 | - | - | - | **3.01** | - | **2.60** |
| **Acc.** ↓ | 8.69 | **1.56** | 6.95 | 19.40 | 2.85 | 2.67 | 5.50 | 3.65 | 2.77 | 3.20 | **2.59** |
| **Comp.** ↓ | 12.12 | 3.33 | 5.33 | 10.19 | 3.00 | 4.55 | 12.29 | 4.16 | **2.45** | 2.39 | **2.28** |
| **Ratio** ↑ | 67.62 | 87.61 | 66.60 | 72.96 | 89.33 | 86.59 | 63.62 | 79.37 | **92.79** | 92.99 | **93.38** |

as the latest methods. Specifically, we report comparisons on all 8 scenes in Replica. As for ScanNet, we report comparison on scene 50, 84, 580, and 616 used by MonoSDF [88], and also scene 59, 106, 169, and 207 used by NICE-SLAM [97]. We mainly report average results over the dataset, please refer to our supplementary materials for results on each scene.

**Metrics.** With the learned occupancy function $f$, we reconstruct the surface of a scene by extracting the zero level set of $f$ using the marching cubes algorithm [40]. Following previous studies [97, 22], we use depth L1 [cm], accuracy [cm], completion [cm], and completion ratio [<5cm%] as metrics to evaluate reconstruction accuracy on Replica. Additionally, we report Chamfer distance (L1), precision, recall, and F-score with a threshold of $0.05$ to evaluate reconstruction accuracy on ScanNet. To evaluate the accuracy in camera tracking, we use ATE RMSE [65] as a metric. We follow the same parameter setting in these metrics as [97].

## 4.2 Evaluations

**Evaluations on Replica.** We use the same set of RGBD images as [66, 96]. We report evaluations in surface reconstruction and camera tracking in Tab. 1, Tab. 2 and Tab. 3, respectively.

We jointly optimize camera poses and learn geometry in the context of SLAM. The depth fusion prior $G_s$ incrementally fuses depth images using estimated camera poses. We report the accuracy of reconstruction from $G_s$ as "TSDF". Compared to this baseline, we can see that our method improves the reconstruction using the geometry learned through volume rendering and occupancy prediction priors. We visualize the advantage of learning geometry in Fig. 2. Note that the holes in TSDF are caused by the absence of RGBD scans. We can see that our neural implicit keeps the correct structure in TSDF and plausibly completes the missing structures in TSDF. We further visualize the attention weights $\beta$ learned for depth fusion in Fig. 3. We visualize the cross sections on 4 scenes, where $\beta$ learned in bandwidth are shown in color red. Generally, the neural implicit mostly focuses

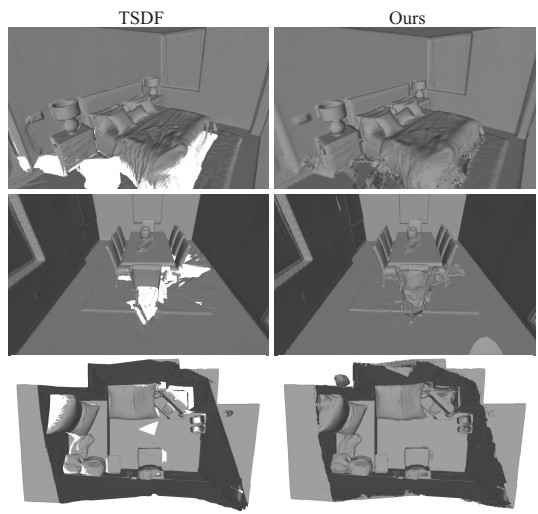

Figure 2: Merits of attentive depth fusion prior.

more on the depth fusion priors in areas where TSDF is complete, while focusing more on the learned geometry in areas where TSDF is incomplete. In the area where TSDF is complete, the network also pays some attention to the inferred occupancy because the occupancy interpolated from TSDF may not be accurate, especially on the most front of surfaces in Fig. 3.

Moreover, our method outperforms the latest implicit-based SLAM methods like NICE-SLAM [97] and NICER-SLAM [96]. We present visual comparisons in Fig. 6, where our method produces more accurate and compact geometry. For method using GT camera poses like vMAP [29] and MonoSDF [88], we achieve much better performance as shown by "Ours*" in Tab. 1 and "Ours" in Tab. 2. Although we require the absolute dense depth maps, compared to MonoSDF [88] that can work with

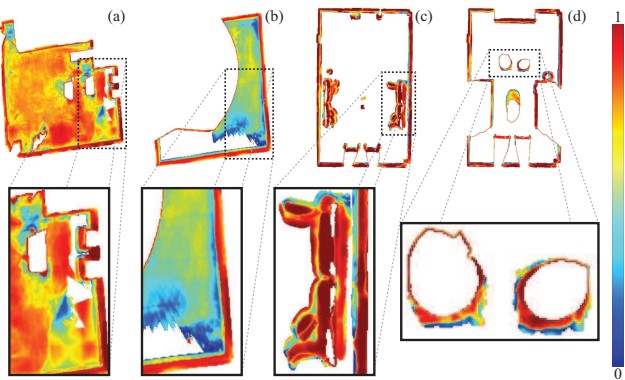

Figure 3: Visualization of attention on depth fusion.

Table 2: Reconstruction Comparison with MonoSDF on Replica.

| | Test Split | | | Train Split | | |
|---|---|---|---|---|---|---|
| | Normal C.↑ | Chamfer-$L_1$ ↓ | F-score ↑ | Normal C.↑ | Chamfer-$L_1$ ↓ | F-score ↑ |
| MonoSDF [88] | 90.56 | 4.26 | 76.42 | **91.80** | 3.59 | 85.67 |
| Ours | **90.69** | **2.43** | **92.47** | 91.05 | **2.73** | **90.52** |

Table 3: Camera Tracking Comparisons (ATE RMSE) on Replica.

| | rm-0 | rm-1 | rm-2 | off-0 | off-1 | off-2 | off-3 | off-4 | Avg. |
|---|---|---|---|---|---|---|---|---|---|
| NICE-SLAM [97] | 1.69 | 2.04 | 1.55 | **0.99** | **0.90** | 1.39 | 3.97 | 3.08 | 1.95 |
| NICER-SLAM [96] | **1.36** | 1.60 | **1.14** | 2.12 | 3.23 | 2.12 | **1.42** | 2.01 | 1.88 |
| Ours | 1.39 | **1.55** | 2.60 | 1.09 | 1.23 | 1.61 | 3.61 | **1.42** | **1.81** |

scaled depth maps, our method only can use the views in front of the current time step, where MonoSDF [88] can utilize all views during the whole training process.

In camera tracking, our results in Tab. 3 achieve the best in average. We compare the estimated trajectories on scene off-4 in Fig. 4 (c). The comparison shows that our attentive depth fusion prior can also improve the camera tracking performance through volume rendering.

**Evaluations on ScanNet.** We further evaluate our method on ScanNet. For surface reconstruction, we compare with the latest methods for learning

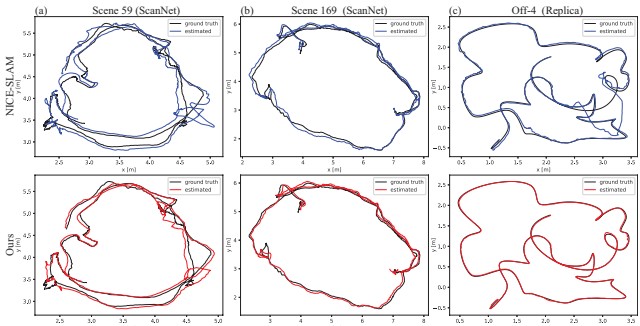

Figure 4: Visual comparisons in camera tracking.

neural implicit from multiview images. We report both their results and ours with GT camera poses, where we also fuse every 10 depth images into the TSDF $G_s$ and render every 10 frames for fair comparisons. Numerical comparisons in Tab. 4 show that our method achieves the best performance in terms of all metrics, where we use the culling strategy introduced in MonoSDF [88] to clean the reconstructed mesh. We highlight our significant improvements in visual comparisons in Fig. 7. We see that our method can reconstruct sharper, more compact and detailed surfaces than other methods. We detail our comparisons on every scene with the top results reported by GO-Surf [75] and NICE-SLAM [97] in Tab. 5 and Tab. 6. The comparisons in Tab. 5 show that our method achieves higher reconstruction accuracy while GO-Surf produces more complete surfaces. We present visual comparisons with error maps in Fig. 5.

In camera tracking, we compare with the latest methods. We incrementally fuse the most current depth frames into TSDF $G_s$ which is used for attentive depth fusion priors in volume rendering. Numerical comparisons in Tab. 7 show that our results are better on 2 out of 4 scenes and achieve the best in average. Tracking trajectory comparisons in Fig. 4 (a) and (b) also show our superiority.

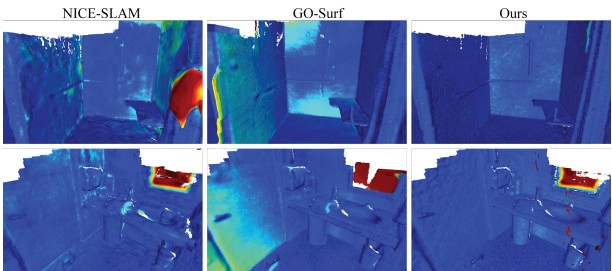

Figure 5: Visual comparison of Error maps (Red: Large).

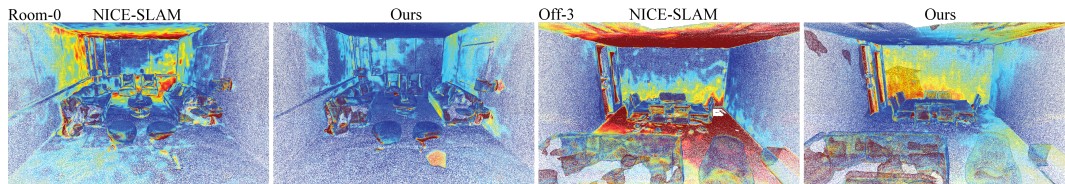

Figure 6: Visual comparisons of error maps (Red: Large) in surface reconstructions on Replica.

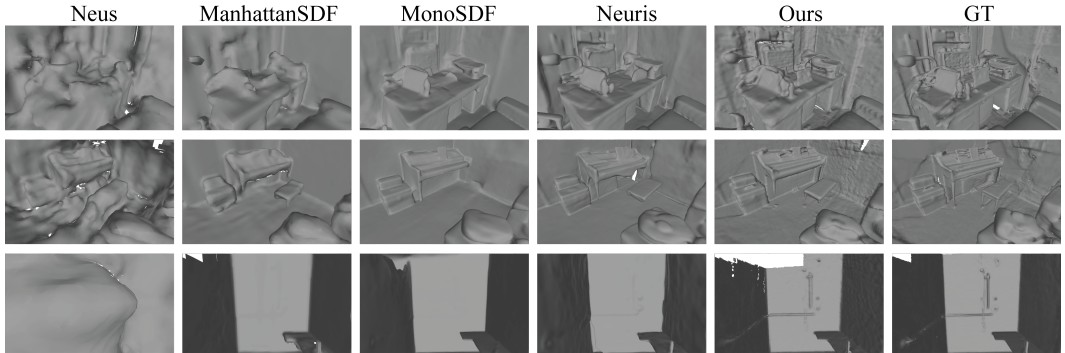

| Neus | ManhattanSDF | MonoSDF | Neuris | Ours | GT |

Figure 7: Visual comparisons in surface reconstructions on ScanNet.

Table 4: Reconstruction Comparisons on ScanNet.

| | Acc ↓ | Comp ↓ | Chamfer-$L_1$ ↓ | Prec ↑ | Recall ↑ | F-score ↑ |
|---|---|---|---|---|---|---|
| COLMAP [62] | 0.047 | 0.235 | 0.141 | 0.711 | 0.441 | 0.537 |
| UNISURF [51] | 0.554 | 0.164 | 0.359 | 0.212 | 0.362 | 0.267 |
| NeuS [77] | 0.179 | 0.208 | 0.194 | 0.313 | 0.275 | 0.291 |
| VolSDF [85] | 0.414 | 0.120 | 0.267 | 0.321 | 0.394 | 0.346 |
| Manhattan-SDF [16] | 0.072 | 0.068 | 0.070 | 0.621 | 0.586 | 0.602 |
| NeuRIS [76] | 0.050 | 0.049 | 0.050 | 0.717 | 0.669 | 0.692 |
| MonoSDF [88] | 0.035 | 0.048 | 0.042 | 0.799 | 0.681 | 0.733 |
| Ours | **0.034** | **0.039** | **0.037** | **0.913** | **0.894** | **0.902** |

Table 5: Reconstruction Comparison with GO-Surf on ScanNet.

| | GO-Surf [75] | | | | | Ours | | | | |
|---|---|---|---|---|---|---|---|---|---|---|
| Scene ID | 0050 | 0084 | 0580 | 0616 | Avg. | 0050 | 0084 | 0580 | 0616 | Avg. |
| Acc ↓ | 0.056 | 0.073 | 0.057 | **0.026** | 0.053 | **0.030** | **0.039** | **0.041** | **0.026** | **0.034** |
| Comp ↓ | **0.024** | 0.017 | **0.024** | 0.023 | **0.022** | 0.043 | **0.014** | 0.035 | 0.063 | 0.039 |
| Chamfer-$L_1$ ↓ | 0.040 | 0.045 | 0.040 | **0.025** | 0.038 | **0.037** | **0.026** | **0.038** | 0.045 | **0.037** |

Table 6: Reconstruction Comparison with NICE-SLAM on ScanNet.

| | NICE [97] | | | | | Ours | | | | |
|---|---|---|---|---|---|---|---|---|---|---|
| Scene ID | 0050 | 0084 | 0580 | 0616 | Avg. | 0050 | 0084 | 0580 | 0616 | Avg. |
| Acc ↓ | **0.030** | **0.031** | **0.032** | **0.026** | **0.030** | **0.030** | 0.039 | 0.041 | **0.026** | 0.034 |
| Comp ↓ | 0.053 | 0.020 | **0.031** | 0.076 | 0.045 | **0.043** | **0.014** | 0.035 | **0.063** | **0.039** |
| Chamfer-$L_1$ ↓ | 0.041 | **0.025** | 0.032 | 0.051 | **0.037** | **0.037** | 0.026 | 0.038 | **0.045** | **0.037** |

### 4.3 Ablation Studies

We report ablation studies to justify the effectiveness of modules in our method on Replica. We use estimated camera poses in ablation studies.

**Effect of Depth Fusion.** We first explore the effect of different ways of depth fusion on the performance. According to whether we use GT camera poses or incremental fusion, we try 4 alternatives with $G_s$ obtained by fusing all depth images at the very beginning using GT camera poses (Offline) or fusing the current depth image incrementally (Online), and using tracking to estimate camera poses (Tracking) or GT camera poses (GT) in Tab. 8. The comparisons show that our method can work well with either GT or estimated camera poses and fusing depth either all together or incrementally in a streaming way. Additional conclusion includes that GT camera poses in either depth fusion and rendering do improve the reconstruction accuracy, and the structure fused from more recent frames is more important than the whole structure fused from all depth images.

Table 7: Camera Tracking Comparisons (ATE RMSE) on ScanNet.

| Scene ID | 0059 | 0106 | 0169 | 0207 | Avg. |
|---|---|---|---|---|---|
| iMAP [66] | 32.06 | 17.50 | 70.51 | 11.91 | 33.00 |
| DI [22] | 128.00 | 18.50 | 75.80 | 100.19 | 80.62 |
| NICE [97] | 12.25 | 8.09 | 10.28 | **5.59** | 9.05 |
| CO [74] | 12.29 | 9.57 | **6.62** | 7.13 | **8.90** |
| Ours | **10.50** | **7.48** | 9.31 | 5.67 | **8.24** |

Table 8: Ablation Studies on Depth Fusion.

| | TSDF | Offline+GT | Online+GT | Offline+Tracking | Online+Tracking (Ours) |
|---|---|---|---|---|---|
| **DepthL1 ↓** | 6.56 | 2.79 | **2.73** | 2.75 | 3.01 |
| **Acc. ↓** | **1.56** | 2.62 | 2.62 | 2.82 | 2.77 |
| **Comp. ↓** | 3.33 | 2.45 | **2.36** | 2.40 | 2.45 |
| **Ratio ↑** | 87.61 | 92.78 | 92.79 | **92.97** | 92.79 |

**Effect of Attention.** We explore the effect of attention on the performance in Tab. 9. First of all, we remove the attention mechanism, and observe a severe degeneration in accuracy, which indicates that the attention plays a big role in directly applying depth fusion priors in neural implicit.

Table 9: Ablation Studies on Attention.

| | w/o Attention | Low | Low + High | High (Ours) |
|---|---|---|---|---|
| **DepthL1 ↓** | 1.86 | 2.75 | 2.96 | **1.44** |
| **Acc. ↓** | 2.69 | 2.96 | 3.85 | **2.54** |
| **Comp. ↓** | 2.81 | 3.14 | 2.91 | **2.41** |
| **Ratio ↑** | 91.46 | 89.73 | **93.63** | 93.22 |

Then, we try to apply attention of depth fusion priors on geometries predicted by different features including low frequency $t_l$, high frequency $t_h$, and both of them. The comparisons show that adding details from depth fusion priors onto low frequency surfaces does not improve the performance. More analysis on attention mechanism can be found in Fig. 9.

**Attention Alternatives.** Our preliminary results show that using softmax normalization layer achieves better performance than other alternatives. Since we merely need two attention weights $\alpha$ and $\beta$ to combine the learned geometry and the depth fusion prior, one alternative is to use a sigmoid function to predict one attention weight and use its difference to 1 as another attention weight.

Table 10: Ablation Studies on Attention Alternatives.

| | Coordinates | Sigmoid | Softmax (Ours) |
|---|---|---|---|
| **DepthL1 ↓** | 1.81 | 1.96 | **1.44** |
| **Acc. ↓** | 2.66 | 2.86 | **2.54** |
| **Comp. ↓** | 2.74 | 2.66 | **2.41** |
| **Ratio ↑** | 93.13 | **93.27** | 93.22 |

However, sigmoid can not effectively take advantage of the depth fusion prior. We also try to use coordinates as an input, which pushes the network to learn spatial sensitive attentions. While the reconstructed surfaces turn out to be noisy in Fig. 8. This indicates that our attention learned a general attentive pattern for all locations in the scene. We also report the numerical comparisons in Tab. 10. Please refer to our supplementary materials for more results related to attention alternatives.

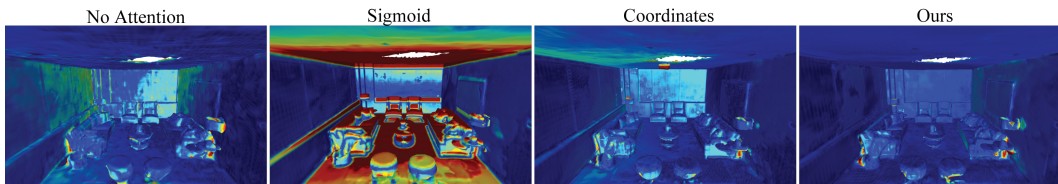

Figure 8: Visual comparison of error maps with different attention alternatives (Red: Large).

**Effect of Bandwidth.** We further study the effect of bandwidth on the performance in Tab. 11. We try to use attentive depth fusion priors everywhere in the scene with no bandwidth. The degenerated results indicate that the truncated area outside the bandwidth does not provide useful structures to improve the performance. Moreover, wider bandwidth may cause more artifacts caused by the calculation of the TSDF $G_s$ while narrower bandwidth brings more incomplete structures, neither of which improves the performance. Instead of using low frequency geometry outside the bandwidth, we also try to use high frequency surfaces. We can see that low frequency geometry is more suitable to describe the scene outside the bandwidth.

Table 11: Ablation Studies on the Effect of Bandwidth.

|  | w/o Bandwidth | Bandwidth=3 | Bandwidth=5 (Ours) | Bandwidth=7 | High | Low (Ours) |
|---|---|---|---|---|---|---|
| **DepthL1** ↓ | 1.87 | 1.61 | **1.44** | 2.01 | 124.21 | **1.44** |
| **Acc.** ↓ | **2.40** | 2.85 | 2.54 | 2.93 | 25.06 | 2.54 |
| **Comp.** ↓ | 3.49 | 2.61 | **2.41** | 2.73 | 3.62 | **2.41** |
| **Ratio** ↑ | 90.94 | 92.84 | 93.22 | **93.55** | 86.92 | **93.22** |

**Effect of Prediction Priors.** Prediction priors from decoders $f_l$ and $f_h$ are also important for accuracy improvements. Instead of using fixed parameters in these decoders, we try to train $f_l$ (Fix $f_h$) or $f_h$ (Fix $f_l$) with other parameters. Numerical comparisons in Tab. 12 show that training $f_l$ or $f_h$ cannot improve the performance, or even fails in camera tracking (Fix $f_h$).

Table 12: Ablation Studies on Prediction Priors.

|  | w/o Fix | Fix $f_l$ | Fix $f_h$ | Fix $f_l + f_h$ (Ours) |
|---|---|---|---|---|
| **DepthL1** ↓ | 125.95 | 43.46 | F | **1.44** |
| **Acc.** ↓ | 127.62 | 67.83 | F | **2.54** |
| **Comp.** ↓ | 56.73 | 18.79 | F | **2.41** |
| **Ratio** ↑ | 5.27 | 103.51 | F | **93.22** |

**More Analysis on How Attention Works.** Additionally, we do a visual analysis on how attention works in Fig. 9. We sample points on the GT mesh, and get the attention weights in Fig. 9 (a). At each point, we show its distance to the mesh from the TSDF in Fig. 9 (b) and the mesh from the inferred occupancy in Fig. 9 (c), respectively. Fig. 9 (d) indicates where the former is smaller than the latter. The high correlation between Fig. 9 (a) and Fig. 9 (d) indicates that the attention network focuses more on the occupancy producing smaller errors to the GT surface. Instead, the red in Fig. 9 (e) indicates where the interpolated occupancy is larger than the inferred occupancy is not correlated to the one in Fig. 9 (a). The uncorrelation indicates that the attention network does not always focus on the larger occupancy input but the one with smaller errors, even without reconstructing surfaces.

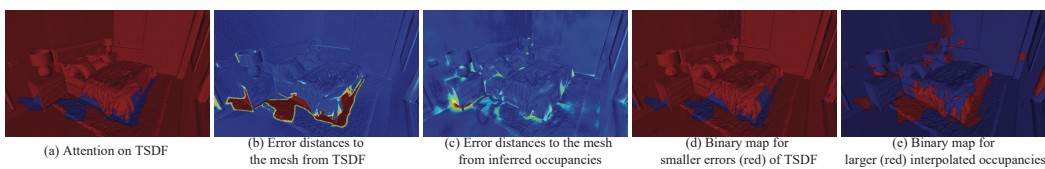

(a) Attention on TSDF  (b) Error distances to the mesh from TSDF  (c) Error distances to the mesh from inferred occupancies  (d) Binary map for smaller errors (red) of TSDF  (e) Binary map for larger (red) interpolated occupancies

Figure 9: Analysis on Attentions. (a) Map the attention weights on occupancies interpolated from TSDF. (b) The point-to-surface distances to the mesh reconstructed from TSDF. (c) The point-to-surface distances to the mesh reconstructed from the inferred occupancies. (d) Binary map indicating smaller errors (red) of TSDF. (e) Binary map indicating larger (red) interpolated occupancies than the inferred occupancies.

# 5  Conclusion

We propose to learn neural implicit through volume rendering with attentive depth fusion priors. Our novel prior alleviates the incomplete depth at holes and the unawareness of occluded structures when using depth images as supervision in volume rendering. We also effectively enable neural networks to determine how much depth fusion prior can be directly used in the neural implicit. Our method can work well with depth fusion from either all depth images together or the ones available in a streaming way, using either known or estimated camera poses. To this end, our novel attention successfully learns how to combine the learned geometry and the depth fusion prior into the neural implicit for more accurate geometry representations. The ablation studies justify the effectiveness of our modules and training strategy. Our experiments on benchmarks with synthetic and real scans show that our method learns more accurate geometry and camera poses than the latest neural implicit methods.

## Acknowledgements and Disclosure of Funding

This work was partially supported by a Richard Barber Interdisciplinary Research Award.

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
