# Supplementary Material for Learning Neural Implicit through Volume Rendering with Attentive Depth Fusion Priors

**Pengchong Hu**     **Zhizhong Han**

Machine Perception Lab, Wayne State University, Detroit, USA

## 1   Implementation Details

In all of our experiments, the voxel size of the low resolution feature grid $G_l$ and the high resolution feature grid $G_h$ is set to 0.32 and 0.16, respectively. The voxel size of the color feature grid $G_c$ is the same as the one in $G_h$. For the TSDF $G_s$, we use a resolution that produces a voxel size of $\frac{1}{64}$. All of the feature vectors in the feature grids have the same dimension $d = 32$.

All MLP decoders have 5 fully-connected blocks, each of which produces a hidden feature dimension of 32. For the pre-trained decoder $f_l$ and $f_h$, we follow the same process and setting as NICE-SLAM [21] during pre-training. Note that the input feature vectors of decoder $f_h$ consist of the interpolated feature vectors from $G_l$ and $G_h$. For the neural function $f_a$, we use an MLP with 6 fully-connected layers, and a Softmax layer that can normalize the output weights $\alpha$ and $\beta$.

For experiments on Replica [10], we shoot $K = 1000$ rays for reconstruction and $K_t = 200$ rays for camera tracking from each view. For experiments on ScanNet [1], we use $K = 5000$ for reconstruction and $K_t = 1000$ for camera tracking from each view. During the optimizing process, the learning rate for optimizing low frequency feature grid $G_l$ is $1e - 1$, for optimizing both low and high frequency feature grid $G_l$ and $G_h$ is $5e - 3$, and for optimizing $G_l$, $G_h$, and $G_c$ jointly is $5e - 3$. The learning rate for tracking on Replica [10] and ScanNet [1] are set to $1e - 3$ and $5e - 3$, respectively. The learning rate for color decoder $f_c$ and neural function $f_a$ are set to $5e - 3$ and $5e - 6$, respectively. For optimizing scene geometry, we use 60 iterations on Replica [10] and ScanNet [1]. For optimizing camera tracking, we use 10 iterations and 50 iterations on Replica [10] and ScanNet [1], respectively.

For experiments in the context of SLAM, we will maintain two TSDF volumes $T$ and $T_{temp}$ for the streaming fusion. After the tracking procedure at time step $t$, the after-fusion stage first fuses the $t$-th depth image into $T$ that has fused all depth images in front using the estimated $t$-th camera pose. Then, we make a copy of $T$ and send it to $T_{temp}$. The pre-fusion stage will fuse the $t + 1$-th depth image into $T_{temp}$ using the camera pose estimated based on constant speed assumption. After that, $T_{temp}$ will be used in tracking procedure to get an updated $t + 1$-th camera pose, with which $T$ can be updated by fusing the $t + 1$-th depth image.

## 2   More Results

Beyond the average results in our paper, we report more detailed results in Tab. 1 and Tab. 2 on Replica [10] and ScanNet [1]. We compare with the latest methods on each scene that we used in evaluations in terms of the same metrics as the previous methods. We can see that our method predicts more accurate geometry than the latest methods on most of scenes. In Tab. 1, we report our results with estimated camera poses as "Ours" and also the results with GT camera poses as "Ours*", where all compared methods are reported with estimated camera poses except for vMAP [5]. Meanwhile, we report our methods with GT camera poses in Tab. 2. The comparisons show that our method can more effectively leverage depth priors to learn neural implicit from RGBD images.

37th Conference on Neural Information Processing Systems (NeurIPS 2023).

Table 1: Reconstruction Comparisons on Replica.

| | | room-0 | room-1 | room-2 | office-0 | office-1 | office-2 | office-3 | office-4 | Avg. |
|---|---|---|---|---|---|---|---|---|---|---|
| **COLMAP** [9] | **Depth L1** [cm] ↓ | - | - | - | - | - | - | - | - | - |
| | **Acc.** [cm] ↓ | 3.87 | 27.29 | 5.41 | 5.21 | 12.69 | 4.28 | 5.29 | 5.45 | 8.69 |
| | **Comp.** [cm] ↓ | 4.78 | 23.90 | 17.42 | 12.98 | 12.35 | 4.96 | 16.17 | 4.41 | 12.12 |
| | **Comp.** Ratio [$< 5$ cm%] ↑ | 83.08 | 22.89 | 64.47 | 72.59 | 69.52 | 81.12 | 64.38 | 82.92 | 67.62 |
| **TSDF-Fusion** [19] | **Depth L1** [cm] ↓ | 4.17 | 6.67 | 6.60 | 3.23 | 4.71 | 11.59 | 9.02 | 6.48 | 6.56 |
| | **Acc.** [cm] ↓ | 1.63 | 1.49 | 1.37 | 1.23 | 1.02 | 2.11 | 2.01 | 1.65 | **1.56** |
| | **Comp.** [cm] ↓ | 3.78 | 3.41 | 3.11 | 1.92 | 2.54 | 3.87 | 3.77 | 4.27 | 3.33 |
| | **Comp.** Ratio [$< 5$ cm%] ↑ | 87.59 | 88.75 | 88.87 | 92.30 | 89.00 | 85.21 | 84.78 | 84.40 | 87.61 |
| **iMAP** [11] | **Depth L1** [cm] ↓ | 5.70 | 4.93 | 6.94 | 6.43 | 7.41 | 14.23 | 8.68 | 6.80 | 7.64 |
| | **Acc.** [cm] ↓ | 5.66 | 5.31 | 5.64 | 7.39 | 11.89 | 8.12 | 5.62 | 5.98 | 6.95 |
| | **Comp.** [cm] ↓ | 5.20 | 5.16 | 5.04 | 4.35 | 5.00 | 6.33 | 5.47 | 6.10 | 5.33 |
| | **Comp.** Ratio [$< 5$ cm%] ↑ | 67.67 | 66.41 | 69.27 | 71.97 | 71.58 | 58.31 | 65.95 | 61.64 | 66.60 |
| **DI-Fusion** [4] | **Depth L1** [cm] ↓ | 6.66 | 96.82 | 36.09 | 7.36 | 5.05 | 13.73 | 11.41 | 9.55 | 23.33 |
| | **Acc.** [cm] ↓ | 1.79 | 49.00 | 26.17 | 70.56 | 1.42 | 2.11 | 2.11 | 2.02 | 19.40 |
| | **Comp.** [cm] ↓ | 3.57 | 39.40 | 17.35 | 3.58 | 2.20 | 4.83 | 4.71 | 5.84 | 10.19 |
| | **Comp.** Ratio [$< 5$ cm%] ↑ | 87.77 | 32.01 | 45.61 | 87.17 | 91.85 | 80.13 | 78.94 | 80.21 | 72.96 |
| **NICE-SLAM** [21] | **Depth L1** [cm] ↓ | 2.11 | 1.68 | 2.90 | 1.83 | 2.46 | 8.92 | 5.93 | 2.38 | 3.53 |
| | **Acc.** [cm] ↓ | 2.73 | 2.58 | 2.65 | 2.26 | 2.50 | 3.82 | 3.50 | 2.77 | 2.85 |
| | **Comp.** [cm] ↓ | 2.87 | 2.47 | 3.00 | 2.02 | 2.36 | 3.57 | 3.83 | 3.84 | 3.00 |
| | **Comp.** Ratio [$< 5$ cm%] ↑ | 90.93 | 92.80 | 89.07 | 94.93 | 92.61 | 85.20 | 82.98 | 86.14 | 89.33 |
| **Vox-Fusion** [16] | **Depth L1** [cm] ↓ | - | - | - | - | - | - | - | - | - |
| | **Acc.** [cm] ↓ | 2.53 | 1.69 | 3.33 | 2.20 | 2.21 | 2.72 | 4.16 | 2.48 | 2.67 |
| | **Comp.** [cm] ↓ | 2.81 | 2.51 | 4.03 | 8.75 | 7.36 | 4.519 | 3.26 | 3.49 | 4.55 |
| | **Comp.** Ratio [$< 5$ cm%] ↑ | 91.52 | 91.34 | 86.78 | 81.99 | 82.03 | 85.45 | 87.13 | 86.53 | 86.59 |
| **DROID-SLAM** [12] | **Depth L1** [cm] ↓ | - | - | - | - | - | - | - | - | - |
| | **Acc.** [cm] ↓ | 12.18 | 8.35 | 3.26 | 3.01 | 2.39 | 5.66 | 4.49 | 4.65 | 5.50 |
| | **Comp.** [cm] ↓ | 8.96 | 6.07 | 16.01 | 16.19 | 16.20 | 15.56 | 9.73 | 9.63 | 12.29 |
| | **Comp.** Ratio [$< 5$ cm%] ↑ | 60.07 | 76.20 | 61.62 | 64.19 | 60.63 | 56.78 | 61.95 | 67.51 | 63.62 |
| **NICER-SLAM** [20] | **Depth L1** [cm] ↓ | - | - | - | - | - | - | - | - | - |
| | **Acc.** [cm] ↓ | 2.53 | 3.93 | 3.40 | 5.49 | 3.45 | 4.02 | 3.34 | 3.03 | 3.65 |
| | **Comp.** [cm] ↓ | 3.04 | 4.10 | 3.42 | 6.09 | 4.42 | 4.29 | 4.03 | 3.87 | 4.16 |
| | **Comp.** Ratio [$< 5$ cm%] ↑ | 88.75 | 76.61 | 86.10 | 65.19 | 77.84 | 74.51 | 82.01 | 83.98 | 79.37 |
| **Ours** | **Depth L1** [cm] ↓ | 1.44 | 1.90 | 2.75 | 1.43 | 2.03 | 7.73 | 4.81 | 1.99 | **3.01** |
| | **Acc.** [cm] ↓ | 2.54 | 2.70 | 2.25 | 2.14 | 2.80 | 3.58 | 3.46 | 2.68 | 2.77 |
| | **Comp.** [cm] ↓ | 2.41 | 2.26 | 2.46 | 1.76 | 1.94 | 2.56 | 2.93 | 3.27 | **2.45** |
| | **Comp.** Ratio [$< 5$ cm%] ↑ | 93.22 | 94.75 | 93.02 | 96.04 | 94.77 | 91.89 | 90.17 | 88.46 | **92.79** |
| **vMAP** [5] | **Depth L1** [cm] ↓ | - | - | - | - | - | - | - | - | - |
| | **Acc.** [cm] ↓ | 2.77 | 3.87 | 1.83 | 4.82 | 3.51 | 3.35 | 3.19 | 2.26 | 3.20 |
| | **Comp.** [cm] ↓ | 1.99 | 1.81 | 2.00 | 3.65 | 2.14 | 2.45 | 2.49 | 2.56 | 2.39 |
| | **Comp.** Ratio [$< 5$ cm%] ↑ | 97.10 | 96.59 | 95.72 | 87.53 | 85.08 | 94.70 | 93.65 | 93.56 | 92.99 |
| **Ours*** | **Depth L1** [cm] ↓ | 1.05 | 0.91 | 1.54 | 0.91 | 1.37 | 8.21 | 5.52 | 1.25 | **2.60** |
| | **Acc.** [cm] ↓ | 2.59 | 2.27 | 2.03 | 2.33 | 2.56 | 3.32 | 3.23 | 2.42 | **2.59** |
| | **Comp.** [cm] ↓ | 2.41 | 1.89 | 2.00 | 1.49 | 1.78 | 2.51 | 3.08 | 3.09 | **2.28** |
| | **Comp.** Ratio [$< 5$ cm%] ↑ | 93.45 | 94.87 | 94.51 | 96.88 | 94.55 | 92.10 | 90.78 | 89.88 | **93.38** |

One thing about the fairness that is worth mentioning in the comparisons is that we follow the SLAM setting and regard the images as a view sequence and only use images that are in front of the current view to infer the neural implicit although we know GT camera poses in Tab. 2. While other methods including UNISURF [8], NeuS [15], VolSDF [17], MonoSDF [18], GO-Surf [13] can use all images at the same time. The information difference makes our method not able to observe the whole scene at the same time. But our attentive depth prior alleviates this demerit, which still leads us to produce better results than the latest methods requiring all images to infer the implicit scene representations.

Additionally, for fair comparisons with MonoSDF [18], we also use GT depth maps to report their results on ScanNet in Tab. 3 and Fig. 3. However, the improvement from GT depth maps is marginal, which is still not as good as ours. We did intend to use the estimated depth images to produce our results. However, we found each estimated depth image used by MonoSDF [18] needs a pair of scale and shift parameters to get normalized, which aligns the estimated point cloud to the scene surface. However, the scale and shift parameters are not consistent across different views, which makes it hard to fuse the estimated depth images into a plausible TSDF, even if using GT camera poses. Fig. 1 shows that the TSDF fails to represent a coarse structure of the scene, which can not be used as a depth fusion prior in our method. Meanwhile, compared to FastSurf [6] that directly uses the TSDF as supervision and can only work in multi- view reconstruction but not SLAM, we report better results in Tab. 4 and Fig. 4.

We also report visual comparisons with data-driven or hole filling methods such as SG-NN [2] and Filling Holes in Meshes [7] in Fig. 2 in the rebuttal. SG-NN fails to fill holes in the scene with ceilings, and [7] produces severe artifacts in empty space due to its limited ability of perceiving the context.

Table 2: Reconstruction Comparisons on ScanNet.

| | | Acc ↓ | Comp ↓ | Chamfer-$L_1$ ↓ | Prec ↑ | Recall ↑ | F-score ↑ |
|---|---|---|---|---|---|---|---|
| scene 0050 | COLMAP [9] | 0.059 | 0.174 | 0.117 | 0.659 | 0.491 | 0.563 |
| | UNISURF [8] | 0.485 | 0.102 | 0.294 | 0.258 | 0.432 | 0.323 |
| | NeuS [15] | 0.130 | 0.115 | 0.123 | 0.441 | 0.406 | 0.423 |
| | VolSDF [17] | 0.092 | 0.079 | 0.086 | 0.512 | 0.544 | 0.527 |
| | Manhattan-SDF [3] | 0.058 | 0.059 | 0.059 | 0.707 | 0.642 | 0.673 |
| | NeuRIS [14] | - | - | - | - | - | - |
| | MonoSDF [18] | - | - | - | - | - | - |
| | GO-Surf [13] | 0.056 | 0.024 | 0.040 | 0.911 | 0.919 | 0.915 |
| | NICE-SLAM [21] | 0.030 | 0.053 | 0.041 | 0.930 | 0.816 | 0.869 |
| | Ours | 0.030 | 0.043 | 0.037 | 0.958 | 0.898 | 0.927 |
| scene 0084 | COLMAP [9] | 0.042 | 0.134 | 0.088 | 0.736 | 0.552 | 0.631 |
| | UNISURF [8] | 0.638 | 0.247 | 0.762 | 0.189 | 0.326 | 0.239 |
| | NeuS [15] | 0.255 | 0.360 | 0.308 | 0.128 | 0.084 | 0.101 |
| | VolSDF [17] | 0.551 | 0.162 | 0.357 | 0.127 | 0.232 | 0.164 |
| | Manhattan-SDF [3] | 0.055 | 0.053 | 0.054 | 0.639 | 0.621 | 0.630 |
| | NeuRIS [14] | - | - | - | - | - | - |
| | MonoSDF [18] | - | - | - | - | - | - |
| | GO-Surf [13] | 0.073 | 0.017 | 0.045 | 0.931 | 0.981 | 0.955 |
| | NICE-SLAM [21] | 0.031 | 0.020 | 0.025 | 0.945 | 0.929 | 0.937 |
| | Ours | 0.039 | 0.014 | 0.026 | 0.924 | 0.963 | 0.943 |
| scene 0580 | COLMAP [9] | 0.034 | 0.176 | 0.105 | 0.809 | 0.465 | 0.590 |
| | UNISURF [8] | 0.376 | 0.116 | 0.246 | 0.218 | 0.399 | 0.282 |
| | NeuS [15] | 0.161 | 0.215 | 0.188 | 0.413 | 0.327 | 0.365 |
| | VolSDF [17] | 0.091 | 0.088 | 0.090 | 0.529 | 0.540 | 0.534 |
| | Manhattan-SDF [3] | 0.104 | 0.062 | 0.153 | 0.616 | 0.650 | 0.632 |
| | NeuRIS [14] | - | - | - | - | - | - |
| | MonoSDF [18] | - | - | - | - | - | - |
| | GO-Surf [13] | 0.057 | 0.024 | 0.040 | 0.911 | 0.920 | 0.915 |
| | NICE-SLAM [21] | 0.032 | 0.031 | 0.032 | 0.939 | 0.888 | 0.913 |
| | Ours | 0.041 | 0.035 | 0.038 | 0.824 | 0.875 | 0.849 |
| scene 0616 | COLMAP [9] | 0.054 | 0.457 | 0.256 | 0.638 | 0.256 | 0.365 |
| | UNISURF [8] | 0.716 | 0.193 | 0.455 | 0.183 | 0.293 | 0.225 |
| | NeuS [15] | 0.171 | 0.142 | 0.157 | 0.269 | 0.284 | 0.276 |
| | VolSDF [17] | 0.922 | 0.150 | 0.536 | 0.115 | 0.259 | 0.160 |
| | Manhattan-SDF [3] | 0.072 | 0.098 | 0.085 | 0.521 | 0.431 | 0.472 |
| | NeuRIS [14] | - | - | - | - | - | - |
| | MonoSDF [18] | - | - | - | - | - | - |
| | GO-Surf [13] | 0.026 | 0.023 | 0.025 | 0.939 | 0.894 | 0.916 |
| | NICE-SLAM [21] | 0.026 | 0.076 | 0.051 | 0.935 | 0.764 | 0.841 |
| | Ours | 0.026 | 0.063 | 0.045 | 0.945 | 0.840 | 0.889 |

Table 3: Reconstruction Comparisons with MonoSDF on ScanNet (scene 0050).

| | Acc ↓ | Comp ↓ | CD-$L_1$ ↓ | Prec ↑ | Recall ↑ | F-score ↑ |
|---|---|---|---|---|---|---|
| Predict | 0.041 | 0.054 | 0.048 | 0.722 | 0.621 | 0.667 |
| GT | 0.039 | 0.049 | 0.044 | 0.763 | 0.682 | 0.721 |
| Ours | **0.030** | **0.043** | **0.037** | **0.958** | **0.898** | **0.927** |

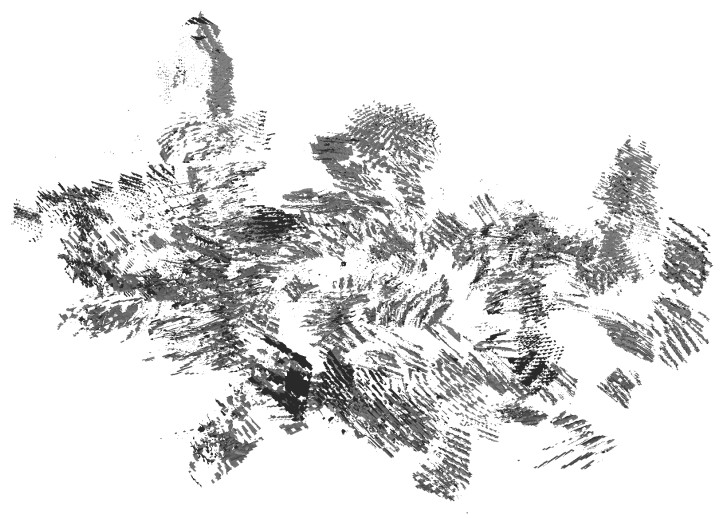

Figure 1: Visualization of TSDF(estimated depth).

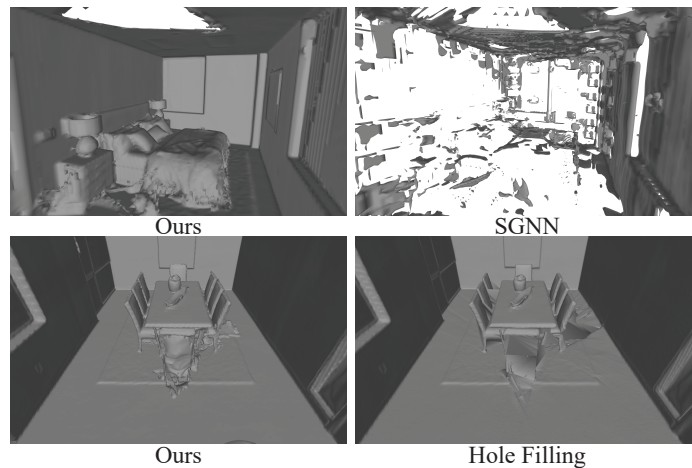

Ours

SGNN

Ours

Hole Filling

Figure 2: Visual comparisons with hole filling methods.

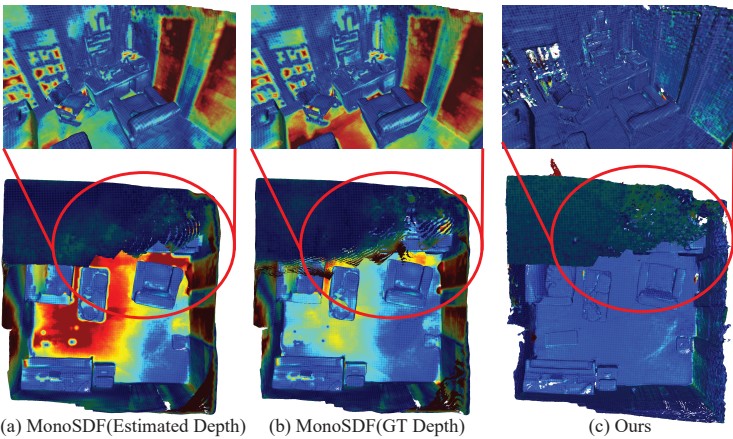

(a) MonoSDF(Estimated Depth)     (b) MonoSDF(GT Depth)     (c) Ours

Figure 3: Visual comparisons of error maps (Red: Large) with MonoSDF.

Table 4: Reconstruction Comparisons with FastSurf on ScanNet.

| Scene ID | | Acc ↓ | Comp ↓ | CD-$L_1$ ↓ | Prec ↑ | Recall ↑ | F-score ↑ |
|---|---|---|---|---|---|---|---|
| 0002 | FS25K | **0.033** | 0.053 | 0.043 | 0.855 | 0.684 | 0.760 |
| | FS75K | **0.033** | 0.057 | 0.046 | 0.819 | 0.655 | 0.728 |
| | Ours | **0.033** | **0.026** | **0.029** | **0.889** | **0.875** | **0.882** |
| 0005 | FS25K | 0.098 | 0.056 | 0.077 | 0.654 | 0.658 | 0.656 |
| | FS75K | 0.099 | 0.057 | 0.088 | 0.621 | 0.622 | 0.621 |
| | Ours | **0.097** | **0.024** | **0.061** | **0.776** | **0.926** | **0.844** |
| 0050 | FS25K | 0.042 | 0.048 | 0.045 | 0.657 | 0.616 | 0.636 |
| | FS75K | 0.042 | 0.048 | 0.045 | 0.670 | 0.625 | 0.647 |
| | Ours | **0.030** | **0.043** | **0.037** | **0.958** | **0.898** | **0.927** |

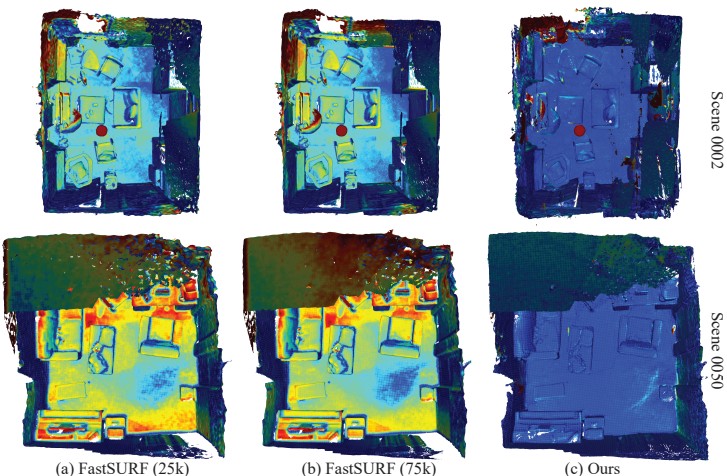

(a) FastSURF (25k)  (b) FastSURF (75k)  (c) Ours

Figure 4: Visual comparisons of error maps (Red: Large) with FastSurf.

Table 5: Ablation Studies on Depth Priors.

| | NICE-SLAM [21] GT+w/o depth loss | Ours GT+w/o depth loss | NICE-SLAM [21] | Ours |
|---|---|---|---|---|
| **Depth L1** [cm] ↓ | 38.11 | **12.82** | 2.11 | **1.44** |
| **Acc.** [cm] ↓ | 18.29 | **8.49** | 2.73 | **2.54** |
| **Comp.** [cm] ↓ | 11.13 | **3.48** | 2.87 | **2.41** |
| **Comp. Ratio** [< 5 cm%] ↑ | 41.47 | **91.35** | 90.93 | **93.22** |

## 3 More Ablation Studies

Beyond the ablation studies in our paper, we report more ablation studies to highlight the effectiveness of using depth fusion priors. As a more effective way of using depth priors than rendering single depth images, we compare the results with or without rendering depth images. Specifically, we use NICE-SLAM [21] as a baseline, and show its results with or without the depth rendering loss during the mapping procedure in Fig. 5. We use the GT camera poses to ensure that inaccurate camera poses do not affect the performance. We keep the experimental setup the same as NICE-SLAM [21], but use our attentive depth prior in our results.

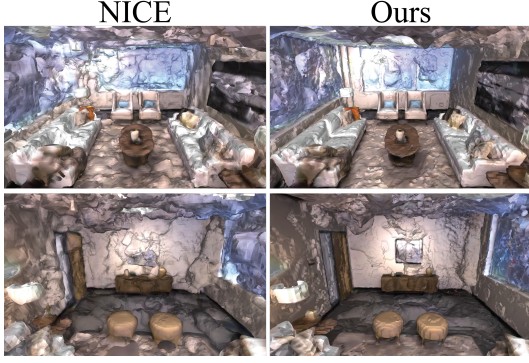

NICE          Ours

Figure 5: Demonstration of the effect of depth loss.

Tab. 5 shows that our attentive depth prior can help network to leverage the depth information with or without using depth rendering loss. Moreover, our attentive depth prior can play a more important

Table 6: Ablation Studies on Attention Alternatives.

|        |                | w/o mlp | max   | Feature | Ours  |
|--------|----------------|---------|-------|---------|-------|
| room0  | **Acc.** ↓     | 2.88    | 2.65  | 2.66    | **2.59** |
|        | **Comp.** ↓    | 2.74    | 2.47  | 2.46    | **2.41** |
|        | **Comp. Ratio** ↑ | 92.70 | 92.84 | 93.39   | **93.45** |
| office0 | **Acc.** ↓    | 2.75    | 2.56  | 2.41    | **2.33** |
|        | **Comp.** ↓    | 2.03    | 1.88  | 1.69    | **1.49** |
|        | **Comp. Ratio** ↑ | 95.236 | 95.57 | 96.10   | **96.88** |

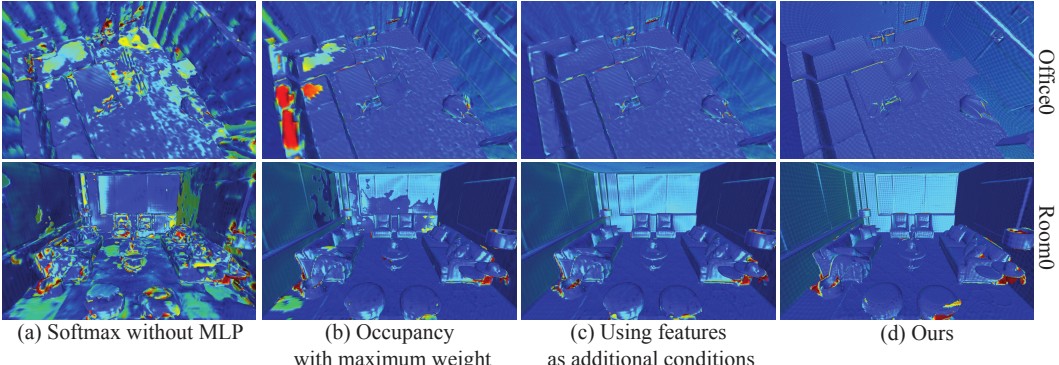

(a) Softmax without MLP     (b) Occupancy with maximum weight     (c) Using features as additional conditions     (d) Ours

Figure 6: Visual comparison of error maps with different attention alternatives (Red: Large).

role to perceive the 3D structure if there is no depth rendering loss used. We also present a visual comparison in Fig. 5 to show the reconstructions of NICE-SLAM [21] and ours without using the depth rendering loss. We can see that our attentive depth priors can significantly improve the reconstruction performance.

Beyond the ablation study about attentive alternatives introduced in main text, we also conduct experiments to report more results with different conditions in Tab. 6 and Fig. 6. All these alternatives degenerate the reconstruction accuracy. Specifically, we remove the MLP and just use a softmax to normalize the two occupancy inputs in Fig. 6(a), use the occupancy with the maximum weight in Fig. 6(b), and use more conditions as input including the features of points that are interpolated from the low and high resolution feature grids in Fig. 6(c).

## 4 More Visualizations

We show more visualizations to present our learning procedure.

**Reconstruction.** First of all, we visualize the learning procedure for reconstruction. We visualize the reconstructed meshes using the occupancy function $f$ learned during training in Fig. 7. To highlight our advantages over NICE-SLAM [21], we also show error maps on reconstructed meshes. We can see that we can learn more accurate implicit functions than NICE-SLAM [21] with our attentive depth fusion prior in different iterations. Please watch our video for more visualization of the reconstruction process.

**Attention.** Then, we visualize the effect of our attention mechanism during our learning procedure. With our attention mechanism, our neural network is able to select better geometry clues at different locations for the learning of implicit representations. In Fig. 8, we visualize the attention weights for the TSDF $G_s$ on a cross section through a scene during training. The attention weights are learned progressively to achieve a stable state so that we can render depth and RGB images that are similar to the ground truth.

**View Rendering.** We compare the rendered RGB and depth images with NICE-SLAM [21] in Fig. 9. The visual comparisons show that our attentive depth fusion prior can also improve the rendering quality. This is also a merit for novel view synthesis.

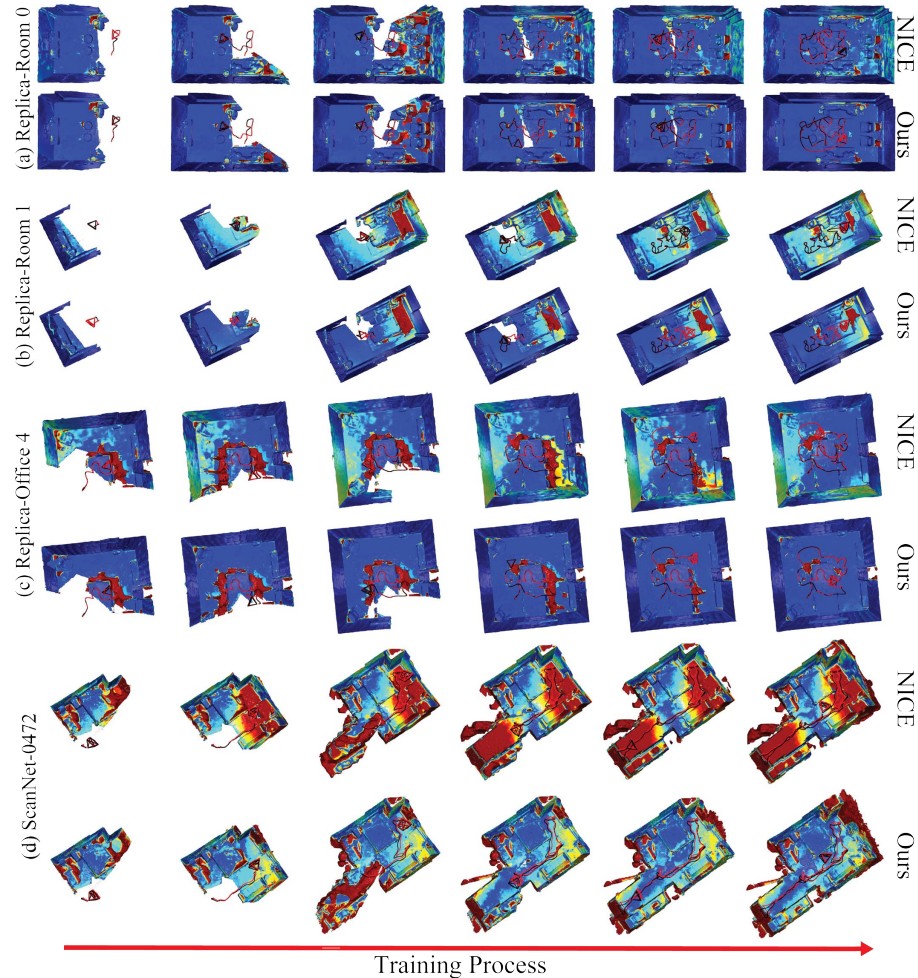

Training Process

Figure 7: Visual comparisons of error maps (Red: Large) during surface reconstructions on Replica and ScanNet.

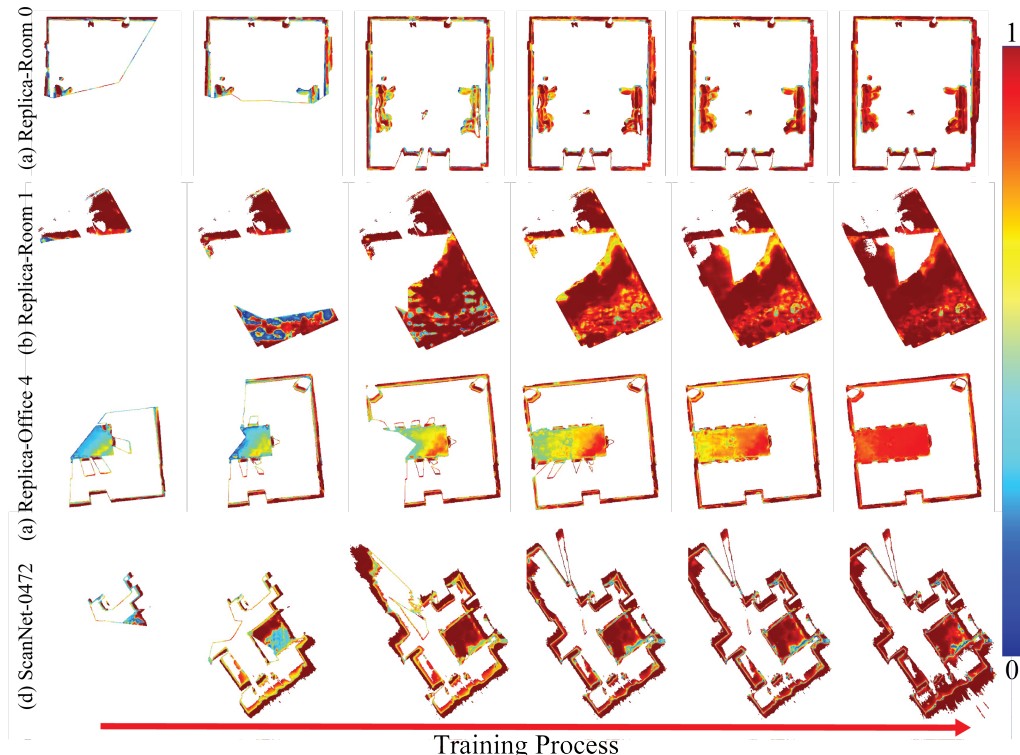

Figure 8: Visualization of attention (Red: Large) on the TSDF $G_s$ during neural implicit inference on Replica and ScanNet.

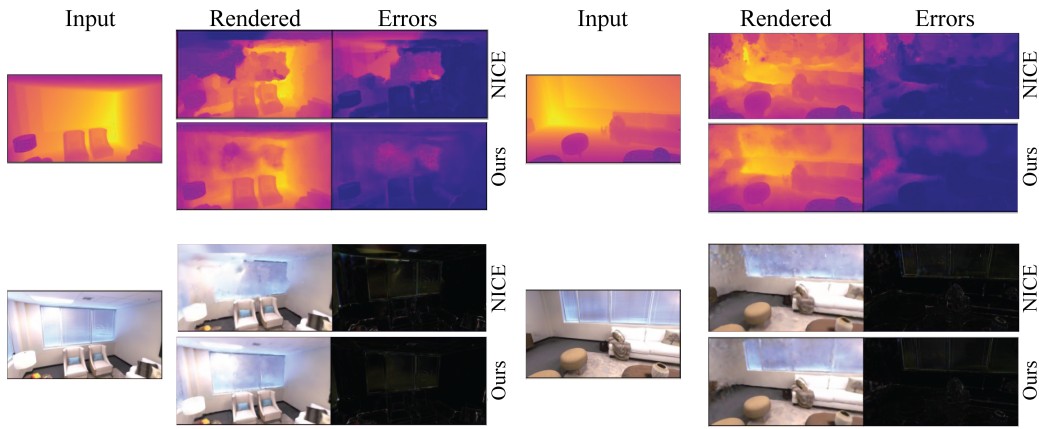

Figure 9: Visual comparisons of rendered images with NICE-SLAM.