# OpenReview forum: "Learning Neural Implicit through Volume Rendering with Attentive Depth Fusion Priors"
_NeurIPS.cc/2023/Conference — NeurIPS 2023 poster_

### Official Review · Reviewer_sZZA · 2023-07-01

**Soundness:** 2 fair
**Presentation:** 2 fair
**Contribution:** 1 poor
**Rating:** 5
**Confidence:** 5

**Summary:**


This paper tackles the problem of 3D scene reconstruction from posed RGB-D images by learning an implicit occupancy function.

Unlike previous methods [58, 3, 65, Ref_DS] that directly use the depth values (obtained either from depth images, SfM pipelines, or from depth prediction networks) of individual rays (i.e. pixels) as supervisory signal leading to learning better scene geometry, faster convergence [Ref_DS, Ref_IN] and inference [Ref_IN, 58], this method proposes to first fuse the depth frames into a TSDF representation using some off-the-shelf method and then query the TSDF for obtaining the occupancy value corresponding to a particular point on a sampled ray for supervision.In other words, the off-the-shelf obtained TSDF of the scene is used as a scene prior.

The motivation for relying on TSDF representation of the scene is that the depth images obtained from RGBD sensors suffer from missing depth values (or holes) which makes it challenging to supervise 3D point samples corresponding to those rays.

The primary contribution of this paper is to learn an attention function that, for a 3d point sample,  estimates the weights for the linear combination of occupancy values which are predicted from the neural network and occupancy values obtained from the TSDF scene prior (as in, final_occupancy = w1*predicted_occupancy + w2*TSDF_obtained_occupancy_prior).  In this sense, the TSDF prior becomes an attentive depth fusion prior.

The network is supervised using volumetric rendering of depth and colour as proposed in [31].

Just like previous methods (e.g. [48, 58]), this method too relies on a hybrid representation: three discrete feature grids are learned, two for low and high-resolution representation of geometry and one for colour, and the tri-linearly interpolated feature value for a corresponding 3D point is decoded by corresponding MLP decoders to obtain occupancy and colour for that 3D point.

The truncated nature of TSDF (it belongs to (-1 , 1) ) is also exploited to inform when to use low-resolution grid features to get occupancy and when to rely on both low and high-resolution grid features to predict occupancy and fuse it (using learned weights) with the TSDF informed occupancy prior; it is clear that points which have TSDF values under the range of 1 or -1 are near surfaces and hence higher resolution feature grids are necessary and in addition, TSDF will explain such points the best.

The method is also applicable for SLAM settings where RGB-D frames are obtained incrementally in which the camera extrinsics, i.e. poses, are also optimized with the network parameters.


[Ref_IN] Kim, Mijeong, Seonguk Seo, and Bohyung Han. "Infonerf: Ray entropy minimization for few-shot neural volume rendering." Proceedings of the IEEE/CVF Conference on Computer Vision and Pattern Recognition. 2022.

[Ref_DS] Deng, Kangle, et al. "Depth-supervised nerf: Fewer views and faster training for free." Proceedings of the IEEE/CVF Conference on Computer Vision and Pattern Recognition. 2022.


**Strengths:**

1. The paper, barring a few places, is well written.

2. This paper has identified the limitation in supervising neural networks that learn implicit functions for dense 3D scene reconstruction using depth images which is difficult in supervising predictions for points whose depth values are not available as they belong to holes in the depth images.

3. Authors have put in a reasonable effort to analyse the proposed method with good qualitative and quantitative analysis.


**Weaknesses:**

1. The paper lacks any substantial contribution

Explanation:

a. It relies on hybrid representation for learning implicit functions for dense 3D reconstruction which has been very well explored before in the same context ([48], [58], [19])

b. The use of pre-trained MLP decoders for predicting occupancy from trilinearly interpolated feature vectors from multi-resolution grids has been inspired by NICE-SLAM[65].

c. The TSDF for the scene is obtained by fusing the posed depth images using the off-the-shelf method.

d. The only contribution of this paper is towards learning an attention function that predicts the weights which are used to linearly fuse the predicted occupancies and occupancy obtained from a TSDF prior.


2. Given the way the attention function is learned which is the primary contribution of this work, it almost looks unnecessary :

Explanation: As explained in the method section and also schematically shown in the pipeline figure (fig 1), the attention function takes the sum of the predicted low and high-resolution grid occupancies and occupancy obtained by TSDF prior (normalized between 0 and 1) to predict the weights for the linear combination of the two (eq. 2, eq 3., figure 1.).

This means the attention function only chooses to learn the weights (alpha and beta, eq. 2, 3) by looking at only two floating point numbers which are between 0 and 1 based on errors in RGB and depth renderings. The RGB and Depth renderings are obtained from the final linear combination of the two occupancies (eq. 2, 3) and their quality would be inversely proportional to the rendering errors.  I don't see what else the attention function can learn to pay attention to other than which one of the inputs is larger.

In which case, how would it be any different from simply using the softmax operator to combine the two occupancies instead of learning two weights?

Had the attention function taken any feature from grids to reason about how well the network can represent the scene , it could make better decision as which is more important, predicted occupancy or TSDF prior occupancy. Accordingly, it would predict weights for the fusion of the two.

3. The method is affected by the very problem it is trying to solve:
Explanation: The paper claims that missing depth values are a problem when using them raw for learning implicit functions for scene reconstruction. The fused TSDF prior would help mitigate that. However, from Figure 2 it is clear that TSDF prior could also have holes where the network has to rely on its own learned occupancy. This shows that the idea of using TSDF is itself affected by the very problem it is trying to mitigate.

4. The paper also proposes that the missing depth due to occlusion leads to difficulty while supervising using depth images. However, when posed RGB-D frames of the scene are available, the points that are occluded in one frame would be visible in some other frames anyway. Consider, for example, the classic NeRF [27] paper or [Ref_DS], the novel view renderings are not affected by occlusions in different frames as the occluded ones are anyway visible in other views.

5. The proposed method is very similar to [19] in the sense that both rely on fused TSDF of the scene then it would be interesting to see how the proposed method fairs with [19]. The qualitative results of [19] look quite similar to the ones shown in this paper which motivates me to also consider a quantitative comparison between the two.


6. The paper points the reviewer to refer to supplemental material for more results. But there is no supplemental material pdf for reference (the supplemental material also has the paper in it).


[Ref_IN] Kim, Mijeong, Seonguk Seo, and Bohyung Han. "Infonerf: Ray entropy minimization for few-shot neural volume rendering." Proceedings of the IEEE/CVF Conference on Computer Vision and Pattern Recognition. 2022.

[Ref_DS] Deng, Kangle, et al. "Depth-supervised nerf: Fewer views and faster training for free." Proceedings of the IEEE/CVF Conference on Computer Vision and Pattern Recognition. 2022.



**Questions:**

1.  How is the proposed attentive depth fusion prior different from simply using a softmax operation over the predicted occupancies and the occupancy obtained from TSDF prior? This question is motivated from the fact that the attention function only takes the two occupancies as input .

2. How is the error in TSDF which is accumulated due to progressive fusion of depth images using their recent estimate of pose handled?

Detail question: In a SLAM setting where we do not have all the RGB-D images beforehand, the TSDF is performed by fusing the scene incrementally using the estimated poses. However, pose estimates from any front-end would have some degree of error. This error would propagate to the TSDF scene prior which is used to construct the final occupancy. The network however is optimized along with camera poses but the TSDF does not seem to be updated based on the optimized camera poses. If, however, it is updated, would it not mean that at every mapping phase, we will have to fuse the entire TSDF again? This would again mean that we will have to keep appropriate number of depth frames for this. However this method, like vMAP[17], iMAP[41], stores keyframes, whose main purpose is for replay while learning incrementally to tackle the catastrophic forgetting issue. No reference to this is provided in the paper.

3. How would the result look like if instead of a linear combination of the predicted occupancy and occupancy from TSDF prior, we use the one with the maximum weight? Could the results indicate that the attention function is redundant? This would also demonstrate if there is any merit in the idea of using TSDF as a prior for composing the occupancies or learning to predict TSDF from TSDF prior is a better idea; [19] has three stages of training, the first is devoted to this.

**Limitations:**

The authors have not discussed any limitations. However, the following are the two strong limitations of this work:

1. One strong limitation of this work is that it would require the availability of TSDF prior even in test time. So what is the point of learning implicitly? The method is also capable of optimizing the camera poses and hence the resulting occupancy would be better (as also shown in the results). But then how is it different from running a classic bundle adjustment and then fusing the optimized depth values into a more accurate TSDF using optimized camera poses? Whereas methods like MonoSDF[58], FastSURF[19] do not need fused TSDF in the test time and the results are quite comparable if not better.

2. Ideally, the attention function could be also provided with the features from the grids so that the function could learn to decide upon the fusion weights also based on the features which are also being optimized. In this process, the learnt features would be considered when deciding the weights required for fusing the occupancies. This would help the network to actually reason about the fusion based on the geometric understanding learned so far.

---

> ### Author Rebuttal · Authors · 2023-08-08
>
> 1. Misunderstanding in summary
>
> * “...a particular point on a sampled ray for supervision. ”
>
> We do not use the interpolated occupancy to supervise the inferred occupancy due to the inaccuracy or incompleteness of TSDF.
>
> * “... TSDF will explain such points the best”
>
> Fig.3 in our manuscript shows that, within bandwidth, TSDF may describe less (such as areas in blue) due to its inaccuracy.
>
> 2. Contributions
>
> Our contribution on the learning framework and the attentive depth fusion prior should get recognized.
>
> One is to show how we should use attention as a balance. Our ablation studies in Tab.7 and Tab.11 showed that it is great to use attention within the bandwidth, and use the inferred low resolution occupancies outside the bandwidth without attention.
>
> The other is attention modeling. Using the interpolated and inferred occupancies without conditions performs the best in ablation studies in Tab.10 and Fig.8. Along with our framework, we believe how we should arrange focus is also worth spreading to the community to benefit the related research.
>
> We did not claim the widely used modules are our contributions.
>
> 3. The attention network
>
> 3.1 Not always focus on the larger occupancy
>
> We report analysis in Fig.1 in the rebuttal. We sample points on the GT mesh, and get the attention weights in Fig.1(a). At each point, we show its distances to the mesh from the TSDF in Fig.1(b) and the mesh from the inferred occupancies in Fig.1(c), respectively. Fig.1(d) indicates where the former is smaller than the later.
>
> The high correlation between Fig.1(a) and Fig.1(d) indicates that the attention network focuses more on the occupancy producing smaller errors to the GT surface.
>
> Instead, the red in Fig.1(e) indicating where the interpolated occupancy is larger than the inferred occupancy is not correlated to the one in Fig.1(a). The uncorrelation indicates that the attention network does not always focus on the larger occupancy input.
>
> 3.2 We used softmax
>
> We would like to point out another reviewer sZZA’s misunderstanding. We did not directly regress two attention weights but used softmax to produce attention weights (L166-167).
>
> 3.3 More attention alternatives
>
> Beyond Fig.8 in the manuscript, we additionally reported more alternatives in Fig.2 in the rebuttal.
>
> * Softmax normalization without MLP.
> * Using the larger occupancy or with maximum weight.
> * Using features as conditions.
>
>
> Comparisons in Fig.2 in the rebuttal show that all these alternatives degenerate the performance.
>
> 4. Holes in Fig.2 do not affect our method
>
> Holes in Fig.2 are invisible areas due to the absence of RGBD scans but not the missing depth values on depth images. We claimed that the TSDF can present missing depth values because of overlaps of neighboring views. However, we did not claim the TSDF can reveal invisible areas.
>
> Moreover, the TSDF does not supervise our implicit representations, so its incompleteness is OK to us. Geometry priors or volume rendering can infer occupancies without TSDF in the invisible area.
>
> 5. Occlusion-awareness is more important in SLAM than view synthesis
>
> We did not claim that the missing depth is caused by occlusion (L33).
>
> Volume rendering relies on multiview consistency constraints to infer occlusion in each single view. Current methods randomly sample rays on different views, randomly sample points on each ray, and expect overlaps among these points to possess multiview consistency constraints in rendering. However, the randomness makes the constraints inefficient to get. It becomes even more difficult in SLAM since one view may only be used once as a rendering target during training if it is not a key frame or far away from the current frame that is being processed, and only views before the current time step can be accessed. However, interpolation on the TSDF can directly estimate coarse occupancy for all points on a ray. It is not affected by the randomness in sampling, consistent across views, and much more occlusion-aware than single depth supervision.
>
>
> 6. Difference to FastSurf [19]
>
> * [19] directly uses the TSDF as supervision.
>
> * [19] can only work in multiview reconstruction but not SLAM.
>
> * [19] can not track camera poses.
>
> * Our advantages are detailed in Fig.4 and Tab.1 in the rebuttal.
>
> 7. Supplemental materials
>
> We are sorry for submitting a wrong supplemental material pdf. We will be happy to provide any important details.
>
> 8. Network difference to simply softmax
>
> As we explained in 3.2, we indeed used softmax as a layer of MLP. Please also see our comparisons in 3.3.
>
> 9. Error accumulation in TSDF
>
> For a tradeoff between the accuracy and the efficiency, we use the most updated camera poses to fuse depth incrementally.
>
> Our solution contains a pre-fusion and an after-fusion stage. After the tracking procedure at timestep t, the pre-fusion stage first fuses the t-th depth image into a TSDF A that fused all depth images in front using the estimated t-th camera pose. This leads to a TSDF B. The mapping procedure uses the TSDF B in learning and also gets an updated t-th camera pose. Then, the after-fusion stage re-fuses the t-th depth image into the TSDF A using the updated t-th camera pose. The new TSDF A will be used at the timestep t+1. We will add a discussion on this in our revision.
>
> 10. Occupancy with the maximum weight
>
> Please see our replies in 3.3. Fig.6 in the rebuttal indicates that the inaccuracy and incompleteness make the TSDF not a good direct supervision in our method.
>
>
> 11. Limitations
>
> 11.1 Depth images during test
>
> Like MonoSDF, FastSurf, and NICE-SLAM, our method is also overfitting-based, all of which need RGBD images to conduct test time optimization (a stage called training but only uses a single test sample). Thus, using RGBD during test time should be a fair evaluation.
>
> Learning implicitly is to learn implicit representations for highly fidelity surface reconstruction.
>
> 11.2 Feature condition for attention
>
> Please see our replies in 3.3.

---

> > ### Comment · Reviewer_sZZA · 2023-08-15
> > **After rebuttle analysis**
> >
> >
> > Response related to misunderstanding of summary:
> >
> > 1)  ì...a particular point on a sampled ray for supervision. î
> > We do not use the interpolated occupancy to supervise the inferred occupancy due to the inaccuracy or incompleteness of TSDF.'
> >
> > ANS: There was no misunderstanding, what was meant from the above line is that TSDF is used for better results. This can be understood from the very next line which states "... used as a scene prior" and not as a supervisory signal.
> > However, I acknowledge the inaccuracy in the pointed out line of my summary.
> >
> > 2) ì... TSDF will explain such points the bestî
> > Fig.3 in our manuscript shows that, within bandwidth, TSDF may describe less (such as areas in blue) due to its inaccuracy.
> >
> > ANS: I would not term the above statement of my review as a misunderstanding.
> > This conclusion was drawn based on the very intent of giving more weightage to in-bandwidth points, as has been stated in L129-130 "Since the TSDF predicts signed distances for queries within the bandwidth with higher confidence than the ones outside the bandwidth, we only use the depth fusion prior within the bandwidth."
> >
> > While the TSDF might describe some parts of such areas inaccurately, its ability to explain the within bandwidth points better is exploited in the paper. Moreover, in Fig 3. , blue and red colors are used to show attention value for "beta" and it is very clear that the attention network is quite rightly giving high values to "beta" for in-bandwidth points.
> >
> > The statement does not tally with claims in the manuscript very well. For example it goes against the very observations the manuscript has provided in L310-313 "We try to use attentive depth fusion priors every where in the scene with no bandwidth. The degenerated results indicate that the truncated area outside the bandwidth does not provide useful structures to improve the performance." From this it is logical to infer that areas within the bandwidth provide useful information to improve performance, which is also the crux of the paper. If not, why even bother to have this distinction between inside bandwidth and outside bandwidth regions and use the in bandwidth points for fusion .
> > The second paragraph of the "Contributions" section of the rebuttal also states the same "...showed that it is great to use attention within the bandwidth, ...".
> >
> > 3) We would like to point out another reviewer sZZAís misunderstanding. We did not directly regress two attention weights but used softmax to produce attention weights (L166-167).
> >
> > Ans: Again there is no misunderstanding here too. I have not suggested anywhere in the review that the paper "regress two attention weights". I think that the authors have confused a rather important question  "how would it be any different from simply using operator to combine the two occupancies instead of learning two weights". By this question, I do not mean that two weights are being learnt. It is very clear in the paper that a softmax is used to produce attenion weights.  What is important here is the question asked.
> >
> > Response with regard to answers provided for my concerns:
> >
> >
> >
> > 1)  my responses to explanation given in "The attention network" segment of the rebuttal:
> > Color coding is not clear in Fig. 1(a). I see only red and blue. However, attention is a real number between 0 and 1. So, I would assume the color coding of Fig. 3 in the paper where red is higher and blue is lower. Looking at Fig 1(a-d), it seems that yes, the attention network does choose the inferred occupancy over the TSDF prior when necessary.  The analysis depicted here answers my Question 1.
> >
> > 2) Answers provided for my concern raised in Weakness 3 of my review is not satisfactory.
> > What difference does it make if the holes in Fig 2. are invisible areas due to absence of RGBD scans or absence of depth values in the depth image, what is important here is the inability of TSDF to help in such a circumstance. This is clearly visible from quality of mesh obtained from inferred occupancies in the those areas shown in Fig 2. So my concern raised in Weakness 3 is not answered from the provided explanation.
> >
> > 3) Concerns raised in Weakness 4  and 5 of my review is addressed and i am satisfied.
> >
> > 4) Limitations 1 and 2 are also well addressed.
> >
> > 5) Concerns related to "Error accumulation in TSDF" during online mode: I am satisfied with the explanation and with the decision of the authors to also put them in the discussion section of their revised manuscript.
> >
> > Overall: The rebuttal answers most of my concerns except for Weakness 3, which I believe is a significant weakness of this work. Nonetheless, I acknowledge that there is some merit in this work and appreciate the authors detailed response which includes running more experiments. I would also request the authors to include the qualitative and quantitative analysis provided in the rebuttal for concerns raised in Question 1,3 and Limitation 2 in the revised manuscript.

---

> > > ### Author Response · Authors · 2023-08-16
> > > **We appreciate your comments and the acceptance rating**
> > >
> > > Dear reviewer sZZA,
> > >
> > > Thanks for your time, comments, and acknowledging our efforts in addressing most of your concerns. We really appreciate that you increased the initial rating to borderline accept.
> > >
> > > Also, we are sorry for any misunderstanding in your review if there was any. We did not mean it.
> > >
> > > 1. A particular point on a sampled ray for supervision
> > >
> > > That is correct, the TSDF is not used as a supervisory signal.
> > >
> > > 2. The TSDF will explain such points the best.
> > >
> > > We would like to further clarify this point and make this point more clear in our revision.
> > >
> > > * Compared to the TSDF outside of the bandwidth, which is merely 1 or -1, we give all credits to the TSDF within the bandwidth (Fig.1), which contains more geometry details, and ignore the TSDF outside the bandwidth. This is also what L129-130 means to say. However, there are no weights involved to balance the TSDF outside the bandwidth and the TSDF within the bandwidth. Thus, saying “giving more weightage to in-bandwidth points” might not be appropriate.
> > >
> > > *  For each point within the bandwidth, the attention network learns to balance its TSDF interpolation within the bandwidth and its occupancy inferred by geometry priors or through volumetric rendering. We acknowledge the dominant role of the TSDF within the bandwidth in L244-246 since the attention network focuses more on the TSDF at most locations (areas in red) on the intersection plane in Fig.3, but obviously not all places. Thus, the areas in blue in Fig.3 do not demonstrate that the TSDF explains such points the best.
> > >
> > > 3. Softmax
> > >
> > > We would like to further clarify this point and make this point more clear in our revision.
> > >
> > > * Yes, we do use softmax to produce the attention weights.
> > > * Regarding “how would it be any different from simply using softmax operator (the original question) to combine the two occupancies instead of two weights”, our reply to this question is the first alternative in 3.3 in our rebuttal. The difference lies in if we use an MLP to manipulate the two occupancy inputs before normalizing them using a softmax. Fig.2 in the rebuttal shows that the MLP is vital to learn the fusion across the scene.
> > >
> > > 4. Color coding in Fig.1(a)
> > >
> > > * The attention weights that we encode in Fig.1 (a) in the rebuttal are real numbers between 0 and 1. The reason why the reviewer merely saw red and blue is that these attention weights are really close to 1 and 0.
> > >
> > > * The ways that we visualize attention weights in Fig. 3 in our paper and Fig. 1 (a) in the rebuttal may determine the difference. Fig. 3 shows attention weights at locations on a randomly selected cross section over the scene while Fig. 1(a) directly shows attention weights at points on the reconstructed mesh at points sampled on the GT mesh . Obviously, almost all points sampled on the mesh are closer to the GT mesh than the ones sampled on the cross section. Thus, you may see more diverse colors in Fig. 3.
> > >
> > > 5. Holes in TSDF
> > >
> > > We would like to further clarify this point and make this point more clear in our revision.
> > >
> > > * In response 4 in our rebuttal, the reason why we explain what causes the holes in Fig. 2 in our paper is a part of our response to your concerns that the TSDF may not be adequate to compensate for the missing depth values on depth images. What we wanted to say is that the TSDF can fill holes caused by noises using neighboring depth images, but it may remain containing holes caused by the absence of RGBD scans.
> > >
> > > * It would not make any difference if there are invisible areas due to the absence of RGBD scans. Our method does not care about holes in the TSDF, or even need to differentiate what caused these holes either, since we have the ability to infer occupancy in invisible areas through geometric priors and volume rendering.
> > >
> > > * Near a hole in TSDF, it is highly possible that this area is outside the bandwidth. This indicates that our method will use the occupancies inferred from the coarse resolution feature grid, and the attention network will not get involved in this area. Thus, the TSDF is not supposed to help in this area.
> > >
> > > * The quality of the reconstructed surface is mainly caused by the challenge conditions in the context of SLAM. We estimate the camera poses, only access the frames in front of each current time step, and incrementally fuse depth images into a TSDF. Please refer to our response 1 Low-quality completion to reviewer QXhf for more details.
> > >
> > >
> > > 6. Revision
> > >
> > > We will follow your instructions to update our paper by adding the numerical and visual comparisons conducted for the analysis in our rebuttal.

---

### Official Review · Reviewer_3uRZ · 2023-07-03

**Soundness:** 4 excellent
**Presentation:** 4 excellent
**Contribution:** 4 excellent
**Rating:** 6
**Confidence:** 5

**Summary:**

This paper proposed a simple yet effective approach of neural implicit surface reconstruction from RGB-D seuqences that leverages the reconstructed TSDF grids as a prior which effectively improve the reconstruction quality of fitting a neaural implicit surface from multi-view RGB-D images directly. The idea of using fused TSDF grids as a geometric prior is novel and makes sense as it comes for free from RGB-D sequences. The proposed attention mechnism of fusing the results from TSDF prior and neural implicit function is also novel and effective through experimental results. The authors have also demonstarted the effectiveness of their proposed methods and each of its components through extensive experiments, ablations and analysis. Except from some very minor issues, this is a good paper. For these reasons, I would reccomend "Weak Accept" at this stage.

**Strengths:**

1. The idea of using fused TSDF grids as a prior is novel. Normally, traditional TSDF fusion could achieve better accuracy in geometry reconstruction but falls short of filling the holes in occluded or unobserved regions. Neual implicit methods, on the other hand, is better at hole filling at the cost of slightly less accurate surfaces. Therefore, using TSDF fusion as a prior and combining these two methods is very natual and has very clear motivation. Also as the authors showed, fused TSDF grid already encoded all historical frames and is occlusion-aware, which could also bring benefits to directly fitting to multi-view RGBD images. Moreover, this prior also comes with no extra cost. The proposed learnable attention module to fuse the output from two sources is also a "simple-yet-effective" solution.

2. Overall, the quantitative experiments are conducted extensively and are sufficient to show the effectiveness of the proposed methods. Quantitative results show the proposed method achieves SOTA  reconstruction and tracking results compared to previous RGB-D neural implicit SLAM/surface reconstruction methods. Qualitative results also look promising and accompanied with detailed and clear analysis. Ablation studies are also well conducted and justified the effectiveness of each module and design choices.

3. The paper is very well written and easy to follow. The introduction and related work sections are very clear and show clear motivation of the proposed method.

4. The method could run both in an online SLAM manner and under batched offline setting, which increases the system's practical value.

**Weaknesses:**

1. Overall, the comparisons with RGB-D baselines seems convincing, but some of the experiments in Sec. 4.2 are comparing against baseline methods that rely on RGB-only input. For example in Tab. 2, 4 and Fig. 5. These are not completely fair comparisons as the proposed methods take measured depth directly. Or did you change some settings in the experiments, like supervising the proposed method with predicted depth instead of measured depth?

2. It would be good to also include performance analysis, such as memory and run-time comparison to previous methods, especially for the SLAM setting.

**Questions:**

Some minor questions:

1. What's the resolution and voxel size used in the feature grids and TSDF grids?

2. Regarding the learnable attention module $f_a(\cdot)$, seems that it is only conditioned on the two occupancy predictions? Have you tried with more conditions?

3. When evaluating reconstruction quality on ScanNet (Tab. 4), what mesh culling strategy did you use? Also is it consistent across all the methods?

4. In the ablation studies on depth fusion (Tab. 8), why would there be difference between GT All and GT? If I understand correctly, the difference is just that GT All operates in a batched manner while GT is online. But this shouldn't cause difference when you run TSDF-fusion?


**Limitations:**

Yes

---

> ### Author Rebuttal · Authors · 2023-08-08
>
> 1. Experimental settings for comparing with multi-view reconstruction methods
>
> Since our method focuses more on SLAM applications which require both camera tracking and mapping two procedures, it is very hard to conduct completely fair comparisons with multi-view reconstruction methods. The comparison details are shown below.
>
> The aspect that we take advantage:
> * Using GT depth images. (while RGB only methods do not use them.)
>
> The aspects that we do not take advantage:
> * Only observe 0-th to (t-1)-th images before the current time step t (while multi-view reconstruction methods can observe all images during the whole training procedure.)
> * Only process frames sequentially (with an interval of 5 frames), and may only see a frame one time during training, if it is not a key frame or far from the current frame that is being processed. (while multi-view reconstruction methods see all images many times during the iterative optimization.)
>
> We did intend to use the estimated depth images to produce our results. However, we found each estimated depth image used by MonoSDF needs a pair of scale and shift parameters to get normalized, which aligns the estimated point cloud to a sparse point cloud obtained by structure from motion. However, the scale and shift parameters are not consistent across different views, which makes it hard to fuse the estimated depth images into a plausible TSDF, even if using GT camera poses. Fig.7 in the rebuttal shows that the TSDF fails to represent a coarse structure of the scene, which can not be used as a depth fusion prior in our method. However, we tried to use GT depth maps and reported their results in Fig.5 and Tab.2 in the rebuttal. As we see the improvement with GT depth images is marginal, which is still not as good as ours.
>
> 2. Computational comparisons
>
> Thanks to the TSDF fusion implemented with CUDA, the runtime of our program(29min56s) is comparable to the one with NICE-SLAM(29min19s) for a scene with 2000 frames. It only takes about 26s to fuse depth images frame by frame incrementally. As for the number of parameters, our method contains 12.15M learnable parameters, which is comparable to the 12.20M ones in NICE-SLAM.
>
> 3. Resolutions
>
> We follow NICE-SLAM and some TSDF fusion algorithms to use different resolutions in different scenes while keeping the size of a voxel the same across different scenes. In a scene, the resolutions of feature grids or TSDF grids are determined by the size of the scene and the size of a voxel. Specifically, the size of a voxel in the low resolution feature grid is 0.32, the size of a voxel in the high resolution feature grid is 0.16, and the size of a voxel in the TSDF grid is 1/64. Using the same size of a voxel aims to generalize the geometry prior learned at the same size of voxel grids.
>
> 4. Attention network alternatives
>
> As mentioned in L165-169, we tried different designs. For instance, without using softmax, we use a sigmoid to predict one weight alpha while using 1-alpha as another weight beta, or use coordinates as a condition. We found that the coordinate condition makes the fusion very sensitive to locations, which degenerates performance. We reported these numerical and visual comparisons in Tab.10 and Fig.8 in the manuscript. We also conduct experiments to explore either the low resolution or high resolution occupancy inferred through volume rendering should be attentive in Tab.11 in the manuscript. Moreover, we conduct experiments to report more results with different conditions in Tab.4 and Fig.2 in the rebuttal, and all these alternatives degenerate the reconstruction accuracy.
>
> 5. Culling strategy
>
> We use the culling strategy introduced in MonoSDF to produce our results and the results of other methods on ScanNet.
>
> 6. GT All vs GT
>
> The difference between GT All and GT in Tab.8 in our manuscript lies in if we fuse all depth maps into a TSDF grid at the very beginning. GT All indicates we do that and use the TSDF containing the whole scene to process all frames. GT indicates that we do not do that but incrementally fuse a depth map at the current time step into the TSDF, which means we can only use part of the scene observed before the current time step. Although the TSDF fusion is the same, how the attention network learns weights to balance the occupancy interpolated from the TSDF and the occupancy inferred through volume rendering is different under these two conditions. Thus, the TSDF difference in learning makes a difference in results.

---

> > ### Comment · Reviewer_3uRZ · 2023-08-19
> >
> > I would like to thank the authors for their detailed response which have addressed most of my concerns. I believe this is a nice paper and there is also enough agreement among reviewers. I will keep my initial positive rating.

---

> > > ### Author Response · Authors · 2023-08-19
> > > **Thanks for your kind words**
> > >
> > > Dear reviewer 3uRZ,
> > >
> > > Thanks for your kind words. We appreciate your effort in reviewing our submission. Your comments are very helpful for us to improve our manucript.
> > >
> > > Best,
> > >
> > > The authors

---

### Official Review · Reviewer_X87y · 2023-07-06

**Soundness:** 3 good
**Presentation:** 3 good
**Contribution:** 3 good
**Rating:** 7
**Confidence:** 4

**Summary:**

The paper proposes a pipeline for estimating scene geometry with TSDF represented with neural implicit function, based on multi-view RGBD inputs. The main novelty is a fusion mechanism which utilized fused depth geometry as prior, and fuses geometry prior with estimated geometry using attention-based weighting. The weighting also considers bandwidth of the TSDF representation, allowing for multi-resolution grid features to be leveraged. The results demonstrated superior performance w.r.t. geometry estimation when compared to methods like MonoSDF, and better pose estimation when compared to methods like NICE-SLAM.

**Strengths:**

[1] The idea to fuse input geometry and estimated geometry, instead of using input geometry as constraints when optimizing estimated geometry, is in general novel and sound. The reasoning is, input geometry from fused depth maps may present high confidence in certain areas, and estimated geometry via MLPs may excel at other regions. By fusing these two instead of estimating all regions with MLPs, the method yields better geometry, and benefits pose estimation in SLAM applications. The insights here would be useful to the community to inspire better ways to leverage input geometry when training MLP-based TSDF fields.

[2] Extensive evaluation. The methods evaluates the method w.r.t. geometry estimation with state-of-the-art methods like MonoSDF, which is the main focus of the paper. Beyond geometry, the method also shows improved geometry benefits downstream tasks of pose estimation, and is able show superior performance over SLAM systems with implicit neural representations. Extensive ablation study is also applaudable, which justifies certain design choices,  as well as providing visualization of what is learned with the attention weights.

[3] The paper is in general well-written and easy to follow.

**Weaknesses:**

[1] Further clarify and justifications are needed in various parts of the paper.

[1.1] For example, on why Transformers are not used to model the attentions, the paper gives an unconvincing reason that people may conjecture that the performance improvements are mainly contributed to Transformers. However, it is unfair and premature to say so without providing actual results with a Transformer-base design. With those additional results, we may then get a better idea on how important the architecture design of the attention mechanism is to the performance gain. Additionally, design choices mentioned in Line 165-169 require justifications via ablation study.

[1.2] Additionally, on the explainability of the learned attention maps, the paper mentions in Line 246 that, 'some area' are paid more attention to, which is not sufficient. Further analysis onto what part of the input geometry is prioritized, and what part is mostly learned, will give better insights of the attention weights, and better justifies the motivation of the proposed. Without further analysis, the attention machoism behaves more or less as a black box.

[1.3] Finally, additional details of the experiment setting are needed. For example, in comparing against MonoSDF, it is not clear whether input depth or estimated depth (with DPT) are used. In experiments where poses are optimized, it is not clear what the initializations look like, what pose representation and regularization are used, and what was the convergence.

[2] Paper writing. For example, the introduction section repeatedly mentions the core contribution of 'attentive depth fusion priors', without giving any explanation or hints of what it exactly is. As a result, the contribution #2 is essentially explaining the 'attentive depth fusion priors', however it is not immediately clear that the 'attentive depth fusion priors' mentioned in contribution #1 is basically explained in #2. Additionally, more discussion should go into explaining import baselines like MonoSDF and NICE-SLAM, on how are they different from the proposed method, so as to better clarify the contributions, and to provide further insights on why the proposed method performs better.

**Questions:**

There are questions included in my reviews, mostly in Weakness 1.2 and 1.3. it would be great if the authors can address those questions.

**Limitations:**

Not applicable. No potential negative societal impacts are mentioned.

---

> ### Author Rebuttal · Authors · 2023-08-03
>
> 1. Attention network justification
>
> We did report ablation studies on attention network justification in our manuscript. We reported numerical and visual comparisons with different attention modeling alternatives in Tab.10 and Fig.8, respectively, where our current attention network produces the best performance. We also reported either the low resolution or high resolution occupancy inferred through volume rendering should be focused on in Tab.11. Additionally, we compared the performance of our simple MLP and the more advanced Transformer network in the rebuttal. Due to the short rebuttal period, we merely saw some results with Transformer that are comparable to ours, which indicates that it still needs more effort to tune the architecture.
>
> 2. Attention analysis
>
> We visualize the learned attention weights on a cross section of the TSDF grid in Fig.3 in the manuscript, and also visualize the attention weights learned at different epochs in our video. Generally, the network mostly focuses more on the occupancy interpolated from the TSDF in areas where TSDF is complete, while focusing more on the occupancy inferred through volume rendering in areas where TSDF is incomplete. In the area where TSDF is complete, the network also pays some attention to the inferred occupancy because the occupancy interpolated from TSDF may not be accurate, especially on the most front of surfaces in Fig.3 in the manuscript.
>
> Moreover, we additionally reported visual analysis on how attention works in Fig.1 in the rebuttal. We sample points on the GT mesh, and get the attention weights in Fig.1(a). At each point, we show its distances to the mesh from the TSDF in Fig.1(b) and the mesh from the inferred occupancies in Fig.1(c), respectively. Fig.1(d) indicates where the former is smaller than the later. The high correlation between Fig.1(a) and Fig.1(d) indicates that the attention network focuses more on the occupancy producing smaller errors to the GT surface. Instead, the red in Fig.1(e) indicating where the interpolated occupancy is larger than the inferred occupancy is not correlated to the one in Fig.1(a). The uncorrelation indicates that the attention network does not always focus on the larger occupancy input.
>
> 3. Experimental details
>
> We are sorry for uploading the wrong supplemental material which includes all the experimental details. Regarding the comparison with MonoSDF, we use the estimated depth maps to report their results. We tried to use GT depth maps and reported their results in Fig.5 and Tab.2 in the rebuttal. But the improvement from GT depth maps is marginal, which is still not as good as ours. Regarding camera tracking, we followed NICE-SLAM and NICER-SLAM to use the GT camera pose for the first frame as the initialization, and also use the same regularization. The camera pose is represented as a 4x4 matrix including the rotation and translation matrices. The optimization is regarded as being converged after a certain amount of iterations in camera tracking and mapping stages respectively. We will upload supplemental materials to report the experimental details.
>
> 4. Paper writing
>
> We will follow your suggestion to revise our manuscript, such as explaining our attentive depth fusion prior and rephrasing our contributions.
>
> 5. Discussions on difference to the latest
>
> We will add more discussions on our advantages over MonoSDF and NICE-SLAM. In summary, our method uses depth fused TSDF as a prior, and employs an attention network to determine where we should use it and how much we should use it with a potential correction from the occupancy inferred through volume rendering. Our attentive depth fusion prior is a more accurate guidance to learn neural implicit representations than using GT depth images as a rendering supervision because single depth images may contain holes and occlusion which can be significantly improved by TSDF. Please see the comparison in Fig.6 in the rebuttal for our advantage over directly using the TSDF as supervision. Moreover, our method also shows novel ways of balancing the confidence within bandwidth and outside bandwidth with attention networks, which also spreads some novel perspectives to use TSDF with neural implicit functions. Our ablation studies in the supplemental material also show that the attentive depth fusion prior can improve surface reconstruction no matter if we are using depth images as rendering targets or not. Hence, the novel way of using depth information differentiates our method from other methods inferring neural implicit representations in either multi-view or SLAM settings, such as MonoSDF and NICE-SLAM.

---

> > ### Comment · Area_Chair_Ewnm · 2023-08-19
> > **Concerns addressed**
> >
> > Thank the authors for their reply. In my opinion, the rebuttal has addressed most of the concerns raised by Reviewer X87y.

---

> > > ### Author Response · Authors · 2023-08-19
> > > **Re: Concerns addressed**
> > >
> > > Dear AC,
> > >
> > > Thank you so much for checking the review and our response. We are so glad to know that our explaination and additional results addressed the reviewer X87y's concerns. We really appreciate your time and expertise.
> > >
> > > Best,
> > >
> > > The authors

---

> > ### Comment · Reviewer_X87y · 2023-08-19
> >
> > I would like to first thank the authors on the rebuttal. As mentioned by the AC, some of my original questions were addressed in the rebuttal with additional results; meanwhile a few more have emerged which I would like to further discuss with the authors. Specifically,
> >
> > [1] On justifying the attention model design. The authors compare the design with a few variants in both the main paper and the rebuttal, which is applaudable. I also understand that it might be impossible to complete experiments with a new Transformer-based design in a short rebuttal window. As such, I stick to my view that comparison with a Transformer-based design is useful to further justify the claims on design choice, and can be added in the final version given time.
> >
> > [2] On the analysis of the attention maps. The authors include in Fig. 1 of the rebuttal more insights into the attention map, which is great. On the analysis by text, the authors emphasize that more weights on TSDF where it is complete, which is intuitively straightforward given that the learned geometry will surely attempt to fill in 'missing holes'. However I wonder more detailed insights will be available. For example, considering the TSDF may present more **high-frequency details** compared to the learned ones, will the learned weights favor more of the TSDF in edges/corners? These are the questions to be answered in a later version of the paper  to provide more insights into what is learned by the method and why it does so.
> >
> > [3] Experiment setting. Thanks to the authors for clarifying on this. However one question has emerged on my side: by using RGBD inputs (with dense GT depth), the experiment setting may not hold for in the wild datasets without dense depth. The goal of MonoSDF is to demonstrate geometry reconstruction WITHOUT input depth WITH ABSOLUTE SCALE, and the proposed method will not be able to do so due to its need for DENSE ABSOLUTE DEPTH for TSDF fusion. Please correct me if I am wrong on this, but if my conjecture is true, the assumption of input depth may undermine the practicablity of the proposed method on in the wild captures with RGB input only. And in this case, the comparison with MonoSDF has to be done by providing MonoSDF with same GT input depth, instead of DPT inferred ones.
> >
> > I will hold my final rating until the authors respond to my comment [3] above/

---

> > > ### Author Response · Authors · 2023-08-20
> > > **Thanks for your questions and comments**
> > >
> > > 1.  Design Justification
> > >
> > > We will follow your advice and report additional comparisons in our revision.
> > >
> > > 2.  Attention Analysis
> > >
> > > We additionally reported visual analysis on how attention works in Fig.1 in the rebuttal. We sample points on the GT mesh, and get the attention weights in Fig.1(a). At each point, we show its distances to the mesh from the TSDF in Fig.1(b) and the mesh from the inferred occupancies in Fig.1(c), respectively. Fig.1(d) indicates where the former is smaller than the later. The high correlation between Fig.1(a) and Fig.1(d) indicates that the attention network focuses more on the occupancy producing smaller errors to the GT surface. Instead, the red in Fig.1(e) indicating where the interpolated occupancy is larger than the inferred occupancy is not correlated to the one in Fig.1(a). The uncorrelation indicates that the attention network does not always focus on the larger occupancy input.
> > >
> > > One important conclusion that we draw from these comparisons currently is that the attention network focuses on the occupancy that produces smaller errors to the GT surface, even without explicitly reconstructing surfaces from occupancies and knowing GT surfaces. Fig.1 (a) in the rebuttal shows that the attention network pays more attention to both low frequency geometry, such as the ground and the wall,  and high frequency geometry, such as the wrinkles on the bed in the TSDF.
> > >
> > > We agree that the TSDF may show more high frequency geometries, but we should also be cautious about the noises and sudden depth changes on depth images, which may make the TSDF not accurate. Attention weights on cross sections in Fig. 3 in our manuscript show that the attention network may focus more on the learned occupancies (areas in blue) on the very front of the surface due to the inaccuracy of the TSDF.
> > >
> > > We would like to conduct more analysis on where the attention network is more interested in the high frequency geometry than the low frequency geometry, and vice versa. We will follow your advice and report these experiments in our revision.
> > >
> > > 3.  Comparison with MonoSDF
> > >
> > > MonoSDF does not require GT depth images, while it indeed requires dense depth at the right scale. Specifically, it employs a monocular depth prior to predict depth images from RGB inputs at first, then it leverages a least-squares criterion to solve for a scale and a shift to align the rendered depth image and the monocular depth image on each view, as shown in Eq. 13 in MonoSDF paper and the compute_scale_and_shift function in the released code at code/model/loss.py. The scale alignment is to make the rendered depth images and the monocular depth images comparable, which results in a dense depth image as a supervision of the rendered image from each view angle. Although MonoSDF manipulates the rendered image to align the absolute scale to the one from the monocular depth prior in Eq. 13 in the paper, our results show that it also works well if we align the scale of monocular images to the absolute scale of rendered images.
> > >
> > > We reported the comparisons with MonoSDF with GT depth images in our response 3 “experimental details” in the rebuttal. The visual and numerical comparisons in Fig. 5 and Tab. 2 in the rebuttal show that the GT depth image brings marginal improvement to MonoSDF, which is still worse than ours.
> > >
> > > We will also add these analysis and experiments in our revision.

---

> > > > ### Comment · Reviewer_X87y · 2023-08-20
> > > >
> > > > I agree that MonoSDF would need to align the estimated depth to compute a scale-invariant depth loss, but this is different than the depth requirement in the proposed method, in that, in order for the TSDF fusion to work, dense depth maps (GT or estimated) have to be of absolute consistent scale. This is a much more strict requirement than in MonoSDF; in fact, per-view depth estimated by DPT will not work at all with the proposed method, given no consistent scale can be estimated.
> > > >
> > > > As a result, in my opinion, the requirement of dense depth maps of absolute is a limitation of the proposed method, but not so with MonoSDF. I would recommend stating this upfront, otherwise the readers may not realize this by simply looking at the comparisons. This limitation would not harm the soundness of the proposed method, but something to be clarified in the method/experiment setting, upfront.
> > > >
> > > > Otherwise the paper and rebuttal look fine to me, and I will stick to my original rating.

---

> > > > > ### Author Response · Authors · 2023-08-21
> > > > > **Thanks for your final rating of acceptance**
> > > > >
> > > > > Dear reviewer X87y,
> > > > >
> > > > > Thanks for your final rating of acceptance.
> > > > >
> > > > > We will follow your advice to elaborate on the difference of depth requirements in our comparisons with MonoSDF in the revision. We will also clarify the experiment setup difference between the multi-view reconstruction in which MonoSDF can use all reviews during the whole training and the SLAM in which our method merely uses views in front of the current time step.
> > > > >
> > > > > Best,
> > > > >
> > > > > Authors

---

### Official Review · Reviewer_QXhf · 2023-07-07

**Soundness:** 3 good
**Presentation:** 3 good
**Contribution:** 2 fair
**Rating:** 5
**Confidence:** 4

**Summary:**

This paper aims to improve the performance of 3D reconstruction from multi-view RGB-D images. The key innovation is the attentive depth fusion prior, which allows the networks to directly use the depth fusion prior with the inferred occupancy as the learned implicit function. Experiments show that the proposed method work for both one-time fused TSDF and incrementally fused TSDF.

**Strengths:**

1. This paper aims to improve 3D reconstruction by better handling cases such as incomplete depth at holes. This is one limitation of existing work and is an important task.

2. The method section is clearly and well explained to help readers understand the details.

3. The authors have included video results that can help better understand the visual result quality of the proposed method.

4. Extensive comparison and ablation studies have been done to show improvements over prior work and justify the decision choices of all components.

**Weaknesses:**

1. L242 claims that the proposed method "plausibly completes the missing structures from TSDF" in Fig. 2. However, the completed surface by the proposed method seems to be of very low-quality. It seems a simple non-learning mesh hole filling algorithm on top of the meshes on the left of Fig. 2 can produce much cleaner results. Learning-based methods [A,B] that predict TSDF for indoor scene completion can also achieve much better results.

[A] Dai et al. SG-NN: Sparse generative neural networks for self-supervised scene completion of rgb-d scans. CVPR 2020.
[B] Dai et al. Scancomplete: Large-scale scene completion and semantic segmentation for 3d scans. CVPR 2018.

2. It is great that the paper has conducted extensive quantitative experiments. However, there are currently 12 tables in the main paper which seems a bit overwhelming. Indeed, some tables are not that critical and can be moved to supplementary. Alternatively, the authors can only present the scene average evaluation in the main paper for some table and move the per-scene results to the supplementary. The new empty space can be instead used to address some limitations.

**Questions:**

1. Since the proposed method is predicting multiple values for the same query points, what is the time increase over a naive neural implicit function (predicting a single signed distance for each query point)?

2. Minor editorial suggestions:
L173: it would be better to swap "color function c" and "occupancy function f" for better correspondence.
L191: includes -> include
L303: attentions -> attention

**Limitations:**

The authors have not adequately addressed the limitations or potential negative societal impact. The authors could show some failures cases that indicates some common patterns where the proposed method would fail on

---

> ### Author Rebuttal · Authors · 2023-08-08
>
> 1. Low-quality completion
>
> As described in L226-228, all completed meshes in the right column in Fig.2 are reconstructed in the context of SLAM. Under this setting, we do camera tracking on each frame, render RGB and depth images every 5 frames for reconstruction by minimizing rendering errors, fuse depth images into TSDF incrementally using the estimated camera poses, and only access 0-th to (t-1)-th GT images at t-th time step. These conditions in SLAM result in a dynamic and incremental context information that is much more incomplete and with more uncertainty than the static and relatively complete context required in scene completion or hole filling algorithms on meshes. This kind of context information determines that our method is only allowed to do an extrapolation-like completion since there is no chance to observe the whole scene. However, data-driven or traditional hole filling methods can access the whole scene and merely do an interpolation-like completion, which makes much difference. As described in L241-242, Fig.2 aims to demonstrate our ability of keeping all the correct structure in TSDF and inferring the missing structure through volume rendering.
>
> Additionally, we report comparisons with data-driven or hole filling methods such as SGNN and Filling Holes in Meshes[SGP03] in Fig.8 in the rebuttal. SGNN fails to fill holes in the scene with ceilings, and [SGP03] produces severe artifacts in empty space due to its limited ability of perceiving the context.
>
> 2. Layout
>
> We will follow your suggestions to improve the layout.
>
> 3. Time cost on more prediction
>
> Actually, we only predict one occupancy at one location, and do not predict multiple ones. Besides the one predicted by our neural network, we interpolated occupancies from the fused TSDF for query points. There is almost no time cost on using trilinear interpolation or the simple attention network. As for the TSDF fusion implemented with CUDA, it merely takes about 25.9s to incrementally fuse all 2000 depth images frame by frame in a scene. For example, in room0 of Replica, the total runtime of our method is 29min56s, which is comparable to 29min19s with NICE-SLAM.
>
> 4. Limitations
>
> Since we do not directly model the lighting in the scene, our method does not work well with transparent  surfaces or surfaces with reflection, such as glass. Reflection may make depth sensors fail to capture depth information or break multi-view consistency on RGB images, both of which degenerate the accuracy of reconstructed surfaces. However, this merely makes our method reconstruct a small area poorly but rather fail to reconstruct a whole scene. Please see Fig.3 in the rebuttal for more details.

---

> > ### Comment · Reviewer_QXhf · 2023-08-17
> >
> > Thanks for the response. After reading all reviewers' comments and the responses from the authors, I am leaning towards keeping my original rating.

---

> > > ### Author Response · Authors · 2023-08-19
> > > **Thanks for the positive rating**
> > >
> > > Dear reviewer QXhf,
> > >
> > > Thanks for the positive rating. We also appreciate your time and comments.
> > >
> > > Best,
> > >
> > > The authors

---

### Author Rebuttal · Authors · 2023-08-09

We thank reviewers for their valuable comments and highlighting our novel and interesting idea (REVA x87y, 3uRZ), extensive evaluations and analysis (REVR QXhf, x87y, 3uRZ, sZZA), well-motivated and sound technical method (REVR QXhf), clear presentation (REVR QXhf), and well-written manuscript (REVR x87y, 3uRZ, sZZA).

We are also sorry for uploading the wrong supplemental material. Thus, reviewers missed some important details about experimental settings, discussions on results, visualizations, and more ablation studies. We will be happy to provide any details in our following discussion period. We will upload the supplemental material if our submission could get accepted.

We respond to each review below and include a rebuttal with tables and figures here.

We will release all reviews, discussion, and our rebuttal even if our paper gets rejected.

---

> ### Author Response · Authors · 2023-08-11
> **We would like to discuss with you**
>
> Dear reviewers,
>
> Thanks for your time and effort to review our submission. We are grateful for your expertise and value your comments.
>
> Please take a look at our response to your questions and our additional tables and figures in the PDF file.
>
> If you have any questions, please feel free to let us know. We are more than happy to discuss our work with you.
>
> Best,
> Authors

---

### Decision · Program_Chairs · 2023-09-21

**Decision:**

Accept (poster)

**Comment:**

This paper proposes a novel method to utilize fused depth geometry as prior for learning-based reconstruction.  The feedback during the rebuttal phase addressed the majority of the concerns raised by the reviewers. Subsequently, the evaluations from all reviewers are largely positive. In agreement with the reviewers' consensus, the AC believes this study is valuable and worth publishing.